# Ensuring Life-long Forgetting in Sequential Unlearning via Source-free Optimization

## Abstract

Machine unlearning has emerged as a crucial area due to increasing privacy and security concerns. However, most existing methods focus on batch unlearning, processing requests in a batch manner, which is impractical for real-world scenarios where unlearning requests can occur at any time. This paper explores a more practical approach, i.e., sequential unlearning, where requests must be processed instantly as they arise. We identify two main challenges with current methods when applied to sequential unlearning, i.e., failure to ensure life-long forgetting, and inefficiency in processing sequential requests. To overcome these challenges, we propose a novel unlearning method tailored for sequential unlearning. Firstly, we incorporate an additional life-long forgetting term into the unlearning objective, and transform risk maximization into minimization to stabilize optimization. Secondly, we establish a source-free optimization by leveraging the loss bound and model parameters. This approach not only avoids the considerable computational costs on retain set, but also eliminates the need for data from past unlearning rounds. Extensive experiments on benchmark datasets demonstrate that our proposed method i) effectively ensures life-long forgetting, ii) maintains the model functionality on the retain set, and iii) exhibits a significant efficiency advantage.

## 1 Introduction

Machine Learning (ML) models have demonstrated significant power, leading to widespread application in various fields (Jumper et al., 2021; OpenAI, 2023). However, concerns about privacy and security have emerged in ML models. On the one hand, privacy regulations, e.g., *right to be forgotten*, mandate that models must be capable of withdrawing individual data upon request (Voigt & Von dem Bussche, 2017). On the other hand, the training data can sometimes contain inaccuracies, harmful content, or biases (Mehrabi et al., 2021). In such cases, it becomes necessary to remove this data to maintain integrity and fairness. These concerns have spurred increasing research into Machine Unlearning (MU) (Bourtoule et al., 2021), where the goal is to enable models to forget specific data (i.e., unlearning target) while retaining the overall model functionality.

Most existing studies on MU assume that all unlearning requests are processed simultaneously or in batches. However, this approach is impractical for real-world applications, where unlearning is a continuous task due to the ever-changing environment and the need to update data content regularly. Consequently, current batch unlearning methods are not suitable for real-world scenarios. In this paper, we focus on the problem of sequential unlearning, which processes a sequence of unlearning requests instantly as they arise in each round. To ensure broad applicability, we assume that no prior information about the requests is available to the unlearning algorithm.

Directly applying batch unlearning methods to sequential unlearning poses two significant challenges. Firstly, sequential unlearning necessitates life-long forgetting, which means ensuring that the data forgotten in any past round remains forgotten throughout the entire sequence. This requirement cannot be met by batch unlearning methods, which focus on unlearning in a single round. As shown in Figure 1, our empirical study demonstrates that data unlearned in earlier rounds can be re-memorized in subsequent rounds. To make matters worse, once data is unlearned, it is permanently deleted from both the model and the database. This means that previously unlearned data cannot be reused to improve the quality of forgetting in subsequent rounds. Secondly, processing each sequential unlearning request individually is inefficient.

For exact unlearning methods (Bourtoule et al., 2021; Yan et al., 2022; Li et al., 2023a), sequential unlearning requires constant retraining, which is computationally prohibitive. Similarly, for approximate unlearning methods (Sekhari et al., 2021; Cha et al., 2024), sequential unlearning repeatedly requires influence estimation on the forget set (often involving Hessian inversion or Fisher matrix) or computations on the retain set (which typically comprises a large portion of the training data).

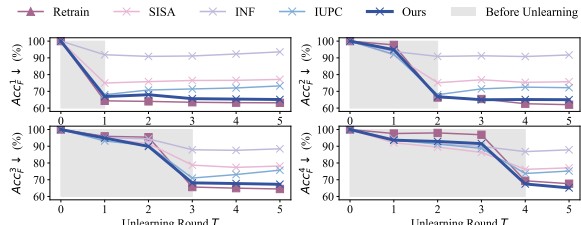

Figure 1: Performance of forget sets in different unlearning rounds, where $Acc_F^t$ denotes the performance on forget set proposed in $t$-th round. The results show that existing methods re-memorize the data unlearned in earlier rounds, while our proposed method can ensure life-long forgetting throughout all rounds. More explanation in Appendix B.1.

To address these two challenges, we propose a novel unlearning method tailored for sequential unlearning. Firstly, we incorporate a life-long forgetting term into the unlearning objective to ensure the consistent forgetting of all past data throughout the following rounds. To stabilize the optimization process, we further transform the loss maximization problem into the minimization problem with pseudo labels. Secondly, we propose a source-free optimization scheme to eliminate the use of both retain set and past unlearned data. As the retain set comprises a large portion of the training data, the repeated computation on retain set in each round costs considerable overhead. Thus, our proposed approach can significantly reduce the computational overhead. Moreover, this approach satisfies the requirement of inaccessible past data. Specifically, our source-free approach utilizes a derived loss bound and the model parameters in previous rounds for optimization. The main contributions of this paper are summarized as follows:

- This paper studies the limitations of batch unlearning in real-world applications. We identify two key challenges and introduce sequential unlearning to address them.
- To address the life-long forgetting challenge, we incorporate an additional term to ensure consistent forgetting of all past data. We further transform this loss maximization term into minimization to stabilize the optimization.
- To address the sequential inefficiency challenge, we propose a source-free optimization scheme that avoids using both the retain set and past unlearned data.
- Extensive experiments on benchmark datasets show that our proposed method i) effectively ensures life-long forgetting, ii) maintains the model functionality on the retain set, and iii) exhibits significant efficiency advantage.

## 2 METHODOLOGY

### 2.1 PROBLEM FORMULATION

In the problem of sequential unlearning, the unlearning requests are raised in chronological order. For conciseness and generality, we assume that the unlearning algorithm has no prior knowledge about unlearning requests, such as the number of data points in each request, the distribution and interdependence of data in requests, and the time intervals between rounds. Thus, the unlearning algorithm treated all requests equally. The arrival time of each request is denoted as $t$-th round with $t \in \mathbb{R}^+$. At the beginning of the unlearning process (i.e., when $t = 0$), the forget set $F_0 = \emptyset$ (i.e., the set of data to be unlearned), while the retain set $R_0$ (i.e., the set of remaining data) is the entire training dataset. At this round, the original model is well-trained on the entire training dataset, denoted as $f_{\boldsymbol{\theta}^0}$, where $\boldsymbol{\theta}$ represents the parameters. At subsequent $t$-th round, a new unlearning request is received, which specifies the forget set $F_t$. Each round of unlearning operates independently, with previous unlearning requests being received and processed instantly. Consequently, the forget set from prior rounds is removed from the dataset, rendering it inaccessible in subsequent rounds, and its memory is considered forgotten. Note that this makes sequential unlearning different from batch unlearning, presenting unique challenges in achieving effective unlearning. After the $t$-th round of unlearning, the model parameters will be updated to $\boldsymbol{\theta}^t$, and the prediction empirical loss of $\boldsymbol{\theta}^t$ given data point $(x, y)$ is denoted as $\ell(f_{\boldsymbol{\theta}^t}(x), y)$.

Figure 2: Overview of the sequential unlearning framework. At $T$-th round, we receive an unlearning request $F_T$, where the total training dataset is divided into three parts: the retrain set $R_T$, the forget set $F_T$, and the combination of forget sets from past rounds (i.e., $\cup_{t=1}^{T-1} F_t$). Among these three parts, only $F_T$ is available to us. With the help of $F_T$, we can compute $\nabla \ell_F^T(\boldsymbol{\theta}^0)$ and $\hat{\boldsymbol{\theta}}_F^T$. Furthermore, under the condition of the state $\nabla g_F^{T-1}$ and the optimal model parameters $\boldsymbol{\theta}_*^{T-1}$ from the previous $T-1$-th round, we sequentially update the state $\nabla g_F^T$ at the current $T$-th round, thereby obtaining the optimal model parameters $\boldsymbol{\theta}_*^T$.

## 2.2 UNLEARNING OBJECTIVE

Following the active unlearning approach (Cha et al., 2024; Kurmanji et al., 2024), the unlearning objective is maximizing the loss on the forget set while minimizing the loss on the retain set. Thus, for sequential unlearning, at $T$-th round, the objective of current unlearning can be formulated as a bi-objective optimization problem:

$$\max_{\boldsymbol{\theta}} \sum_{(x,y) \in F_T} \ell(f_{\boldsymbol{\theta}}(x), y), \min_{\boldsymbol{\theta}} \sum_{(x,y) \in R_T} \ell(f_{\boldsymbol{\theta}}(x), y). \tag{1}$$

According to our empirical results in Figure 1, in the context of sequential unlearning, optimizing Eq. (1) cannot ensure life-long forgetting. In fact, the completeness of unlearning (the quality of forgetting) achieved in previous rounds deteriorates in subsequent rounds.

To ensure life-long forgetting, we need to minimize the cumulative empirical risk in sequential unlearning. Thus, Eq. (1) can be reformulated as the following form:

$$\max_{\boldsymbol{\theta}} \sum_{(x,y) \in \cup_{t=1}^{T-1} F_t} \ell(f_{\boldsymbol{\theta}}(x), y) + \sum_{(x,y) \in F_T} \ell(f_{\boldsymbol{\theta}}(x), y), \tag{2}$$

$$\min_{\boldsymbol{\theta}} \sum_{(x,y) \in R_T} \ell(f_{\boldsymbol{\theta}}(x), y). \tag{3}$$

However, this objective function cannot be directly optimized due to three main issues:

- **Issue 1 (Unstable Optimization)**: For Eq. (2), unlearning through maximizing the empirical risk on forget set is not stable, as it fails to converge to a fixed solution. Thus, while it reduces performance on the forget set (i.e., achieving unlearning), it may also significantly diminish the overall utility of the model (Halimi et al., 2022).
- **Issue 2 (Data Inaccessibility)**: In the context of sequential unlearning, at $T$-th round, we cannot access to the forget sets from previous rounds 1 to $T-1$, thus making it impossible to directly compute Eq. (2).
- **Issue 3 (Prohibitive Computation)**: Eq. (3) relies on the retain set. In the context of unlearning, the size of retain set typically significantly exceeds that of the forget set, leading to considerable computational costs. Moreover, legal and regulatory constraints may further restrict frequent access to the retain set in each round.

## 2.3 SOURCE-FREE SEQUENTIAL UNLEARNING FRAMEWORK

To address the aforementioned issues, we propose a source-free sequential unlearning framework. For issue 1, we transform loss maximization into minimization by introducing pseudo labels. As

shown in Figure 2, our proposed sequential unlearning framework requires only the use of current forget set $F_t$, being source-free regarding the previous forget sets $\cup_{i=1}^{t-1}F_i$ and retrain set $R_t$. Therefore, our approach effectively addresses issues 2 and 3. In the remainder of this section, we describe our proposed framework in details.

**Issue 1.** To address the unstable optimization, we avoid maximizing the model's empirical risk on the forget set. Instead, we create pseudo labels $\hat{y}$ and transform the problem into minimizing the cross-entropy loss between the model's output on the forget set and $\hat{y}$. The method for selecting pseudo-labels is not fixed; we will compare different selection strategies in subsequent experiments. Therefore, we rewrite Eq. (2) as:

$$\min_{\boldsymbol{\theta}} \sum_{(x,y)\in\cup_{t=1}^{T-1}F_t} \ell(f_{\boldsymbol{\theta}}(x), \hat{y}) + \sum_{(x,y)\in F_T} \ell(f_{\boldsymbol{\theta}}(x), \hat{y}). \tag{4}$$

To facilitate optimization, adhering to the setup of (Li et al., 2024), we further transform this bi-objective problem into a single-objective problem through weighted addition. Let $\ell_F^t(\boldsymbol{\theta}) = \sum_{(x,y)\in F_t} \ell(f_{\boldsymbol{\theta}}(x), \hat{y})$ and $\ell_R^t(\boldsymbol{\theta}) = \sum_{(x,y)\in R_t} \ell(f_{\boldsymbol{\theta}}(x), y)$, the unlearning objective can be rewritten as:

$$\min_{\boldsymbol{\theta}} \sum_{t=1}^{T-1} \ell_F^t(\boldsymbol{\theta}) + \ell_F^T(\boldsymbol{\theta}) + \alpha\ell_R^T(\boldsymbol{\theta}), \tag{5}$$

where $\alpha$ is a weight that controls the trade-off between unlearning completeness and model utility.

**Issue 2.** The objective of Eq. (5) is to find a $\boldsymbol{\theta}_*^T = \arg\min_{\boldsymbol{\theta}} \sum_{t=1}^{T-1} \ell_F^t(\boldsymbol{\theta}) + \ell_F^T(\boldsymbol{\theta}) + \alpha\ell_R^T(\boldsymbol{\theta})$. However, since we cannot access the forget sets from the 1-st to $T$-th rounds, the direct calculation of Eq. (5) becomes infeasible. Following (Mirzadeh et al., 2020), we approximate each empirical risk term in Eq. (5) by second-order Taylor expansion. Taking $\ell_F^t(\boldsymbol{\theta}_*^T)$ as an example, it can be approximated as

$$\ell_F^t(\boldsymbol{\theta}_*^T) \approx \ell_F^t(\hat{\boldsymbol{\theta}}_F^t) + (\boldsymbol{\theta}_*^T - \hat{\boldsymbol{\theta}}_F^t)^\top \nabla\ell_F^t(\hat{\boldsymbol{\theta}}_F^t) + \frac{1}{2}(\boldsymbol{\theta}_*^T - \hat{\boldsymbol{\theta}}_F^t)^\top \nabla^2\ell_F^t(\hat{\boldsymbol{\theta}}_F^t)(\boldsymbol{\theta}_*^T - \hat{\boldsymbol{\theta}}_F^t)$$

$$\leq \ell_F^t(\hat{\boldsymbol{\theta}}_F^t) + \frac{1}{2}\lambda_F^t \|\boldsymbol{\theta}_*^T - \hat{\boldsymbol{\theta}}_F^t\|^2, \tag{6}$$

where $\hat{\boldsymbol{\theta}}_F^t$ represents the optimal parameters that the model should satisfy for only unlearning the forget set in $t$-th round (i.e., $\hat{\boldsymbol{\theta}}_F^t = \arg\min \ell_F^t(\boldsymbol{\theta})$), with $\lambda_F^t$ and $\lambda_R^t$ being the maximum eigenvalue of $\nabla^2\ell_F^t(\hat{\boldsymbol{\theta}}_F^t)$ and $\nabla^2\ell_R^t(\hat{\boldsymbol{\theta}}_R^t)$ respectively (He et al., 2015; Ghorbani et al., 2019). Follwoing (Koh & Liang, 2017; Koh et al., 2019), since the optimal value is taken at point $\hat{\boldsymbol{\theta}}_F^t$, then we have $\nabla\ell_F^t(\hat{\boldsymbol{\theta}}_F^t) \approx 0$. Sum up the expanded items in Eq. (5), we can obtain:

$$L_T(\boldsymbol{\theta}_*^T) = \sum_{t=1}^{T-1} \ell_F^t(\boldsymbol{\theta}_*^T) + \ell_F^T(\boldsymbol{\theta}_*^T) + \alpha\ell_R^T(\boldsymbol{\theta}_*^T)$$

$$\leq \frac{1}{2}\lambda_{max} \sum_{t=1}^{T} \|\boldsymbol{\theta}_*^T - \hat{\boldsymbol{\theta}}_F^t\|^2 + \frac{\alpha}{2}\lambda_{max}\|\boldsymbol{\theta}_*^T - \hat{\boldsymbol{\theta}}_R^T\|^2 + \sum_{t=1}^{T} \ell_F^t(\hat{\boldsymbol{\theta}}_F^t) + \alpha\ell_R^T(\hat{\boldsymbol{\theta}}_R^T), \tag{7}$$

where $\lambda_{max} = \max(\lambda_F^1, \lambda_F^2, \ldots, \lambda_F^T, \lambda_R^T)$. It is noted that the right-hand side of Eq. equation 7 is a quadratic term w.r.t. $\boldsymbol{\theta}_*^T$; therefore, the weighted linear combination of $\hat{\boldsymbol{\theta}}_F^t$ and $\hat{\boldsymbol{\theta}}_R^T$ can be regarded as an optimal choice for the upper bound of the target loss function $L_T(\boldsymbol{\theta}_*^T)$ in Eq. equation 7, as shown in Eq. equation 8:

$$\boldsymbol{\theta}_*^T = \frac{1}{T+\alpha} \sum_{t=1}^{T} \hat{\boldsymbol{\theta}}_F^t + \frac{\alpha}{T+\alpha}\hat{\boldsymbol{\theta}}_R^T. \tag{8}$$

Assuming that the optimal model parameter $\boldsymbol{\theta}_*^{T-1}$ has been obtained at $T-1$-th round, then based on Eq. (8), the optimal model parameter at $T$-th round can be represented as

$$\boldsymbol{\theta}_*^T = (1 - \frac{1}{T+\alpha})\boldsymbol{\theta}_*^{T-1} + \frac{1}{T+\alpha}\hat{\boldsymbol{\theta}}_F^T + \frac{\alpha}{T+\alpha}(\hat{\boldsymbol{\theta}}_R^T - \hat{\boldsymbol{\theta}}_R^{T-1}). \tag{9}$$

It can be observed that Eq. (9) does not require the forget set from the 1-st to $T-1$-th rounds; it only necessitates having the optimal model parameter from the $T-1$-th round. As a result, we can address the data inaccessibility issue by directly solving Eq. (9).

**Issue 3.** Due to the size (i.e., data volume) of $R_T$ significantly exceeding that of $F_T$ in the unlearning scenarios, the computational costs required to explicitly obtain $\hat{\boldsymbol{\theta}}_R^T$ on $R_T$ are extremely large. This makes calculating the last term $(\hat{\boldsymbol{\theta}}_R^T - \hat{\boldsymbol{\theta}}_R^{T-1})$ in Eq. (9) challenging. To address this issue, we transform this term into the following two parts: $(\hat{\boldsymbol{\theta}}_R^T - \boldsymbol{\theta}^0) - (\hat{\boldsymbol{\theta}}_R^{T-1} - \boldsymbol{\theta}^0)$, where $\boldsymbol{\theta}^0$ are the parameters of the original model $f_{\boldsymbol{\theta}^0}$.

Previous research estimates the impact of data points on model parameters using influence functions (Koh & Liang, 2017; Koh et al., 2019). Inspired by this, we utilize influence functions to calculate the impact of the missing data points on the original model parameters as an approximation for each term:

$$\hat{\boldsymbol{\theta}}_R^T - \boldsymbol{\theta}^0 \approx -\frac{|R_0 \setminus R_T|}{|R_0|} H_{\boldsymbol{\theta}^0}^{-1} \nabla g_R^T(\boldsymbol{\theta}^0), \tag{10}$$

where $|R_0|$ represents the number of data points in the initial dataset $R_0$, $H_{\boldsymbol{\theta}^0} = \frac{1}{|R_0|} \nabla^2 \ell_R^0(\boldsymbol{\theta}^0)$ is the Hessian matrix, and $g_R^T(\boldsymbol{\theta}^0) = \sum_{(x,y) \in R_0 \setminus R_T} \ell(f_{\boldsymbol{\theta}^0}(x), y)$. Since $R_0 \setminus R_T = \cup_{t=1}^T F_t$, the approximation formulas for these two terms combined can be expressed as (proof in Appendix C):

$$\hat{\boldsymbol{\theta}}_R^T - \hat{\boldsymbol{\theta}}_R^{T-1} \approx \frac{|\cup_{t=1}^T F_t|}{|R_0|} H_{\boldsymbol{\theta}^0}^{-1} \nabla \ell_F^T(\boldsymbol{\theta}^0) + \frac{|F_T|}{|R_0|} H_{\boldsymbol{\theta}^0}^{-1} \nabla g_R^{T-1}(\boldsymbol{\theta}^0). \tag{11}$$

Eq. (11) significantly enhances computational efficiency. First, $H_{\boldsymbol{\theta}^0}$ represents the Hessian matrix of all data in the original model $f_{\boldsymbol{\theta}^0}$, which is independent of unlearning round $T$. This means that we only need to compute $H_{\boldsymbol{\theta}^0}$ once at the initial round, store it in memory, and retrieve it directly in the following rounds without recalculating. Secondly, we only need to calculate the gradient of the loss for the forget set $F_T$ in the original model $f_{\boldsymbol{\theta}^0}$ at the $T$-th round, because $\nabla g_R^{T-1}(\boldsymbol{\theta}^0)$ has already been computed in the previous $T-1$-th round.

**Putting Together.** Based on the analysis above, we can conclude that the optimal model parameters $\boldsymbol{\theta}_*^T$ at $T$-th round should satisfy

$$\boldsymbol{\theta}_*^T \approx (1 - \frac{1}{T+\alpha})\boldsymbol{\theta}_*^{T-1} + \frac{\alpha \frac{|F_T|}{|R_0|} H_{\boldsymbol{\theta}^0}^{-1} \nabla g_R^{T-1}(\boldsymbol{\theta}^0)}{T+\alpha} + \frac{1}{T+\alpha}\hat{\boldsymbol{\theta}}_F^T + \frac{\alpha \frac{|\cup_{t=1}^T F_t|}{|R_0|} H_{\boldsymbol{\theta}^0}^{-1} \nabla \ell_F^T(\boldsymbol{\theta}^0)}{T+\alpha}, \tag{12}$$

where both $\boldsymbol{\theta}_*^{T-1}$ and $\nabla g_R^{T-1}(\boldsymbol{\theta}^0)$ originate from the previous $T-1$-th round. Specifically, as shown in Figure 2, at the $T$-th round, we only need to compute the gradient $\nabla \ell_F^T(\boldsymbol{\theta}^0)$, and $\hat{\boldsymbol{\theta}}_F^T$ which is obtained by tuning $\boldsymbol{\theta}^0$ on a small amount of data, i.e., $F_T$. We summarize our proposed framework in Appendix D.1 and the computational implementation in Appendix D.2.

## 3 EXPERIMENTS

### 3.1 EXPERIMENTAL SETTINGS

**Datasets and Models.** We select three benchmark image classification datasets, i.e., CIFAR-10, CIFAR-100 (Krizhevsky et al., 2009), and Fashion-MNISTXiao et al. (2017), for evaluation. We choose three mainstream image classification models, i.e., ResNet-18 (He et al., 2016), VGG (Simonyan & Zisserman, 2014), and VIT (Dosovitskiy et al., 2020). These models are representative in the field of image processing, embodying both convolutional networks and transformer networks.

**Baselines.** We compare our proposed method with four representative baselines that can comply with the constraints of sequential unlearning scenarios, i.e., the inaccessibility of past forget sets and the current retain set. These baseline methods include (implementation details of these methods are provided in Appendix E.1): **Retrain**: Retraining from scratch is a naive method that retrains a new model on the retain set. **SISA** (Bourtoule et al., 2021): SISA is a well-acknowledged exact unlearning method. **INF** (Sekhari et al., 2021): INF representative approximate (reverse) unlearning method based on influence functions. **IUPC** (Cha et al., 2024): IUPC is the state-of-the-art method belonging to the approximate (active) approach.

**Evaluation Metrics.** We employ the following five metrics to evaluate the effectiveness and efficiency of compared unlearning methods. i) **Current forgetting**: Accuracy on the forget set of

Table 1: Results w.r.t. the effectiveness of sequential unlearning. Except for Original and Retrain, the best results are highlighted in **bold**, and secondary results are highlighted with _underline_.

| Model | Method | CIFAR-10 | | | | CIFAR-100 | | | | Fashion-MNIST | | | |
|---|---|---|---|---|---|---|---|---|---|---|---|---|---|
| | | $Acc_F(\downarrow)$ | $Acc_P(\downarrow)$ | $Acc_R(\uparrow)$ | $Acc_T(\uparrow)$ | $Acc_F(\downarrow)$ | $Acc_P(\downarrow)$ | $Acc_R(\uparrow)$ | $Acc_T(\uparrow)$ | $Acc_F(\downarrow)$ | $Acc_P(\downarrow)$ | $Acc_R(\uparrow)$ | $Acc_T(\uparrow)$ |
| ResNet-18 | Original | 95.77 | 95.15 | 96.18 | 90.56 | 96.26 | 95.07 | 96.02 | 88.16 | 97.67 | 97.06 | 98.44 | 90.67 |
| | Retrain | 62.12 | 58.27 | 95.76 | 82.56 | 31.60 | 27.45 | 70.97 | 63.87 | 73.81 | 66.15 | 94.68 | 83.17 |
| | SISA | 76.02 | 77.44 | 87.57 | 74.13 | 55.27 | 57.61 | _68.33_ | 56.59 | 77.85 | 78.70 | 82.87 | 73.19 |
| | INF | 91.73 | 91.86 | **92.85** | 75.37 | 65.07 | 66.32 | 68.06 | _61.71_ | 90.31 | 90.60 | **92.51** | 81.79 |
| | IUPC | _69.87_ | _72.86_ | 85.72 | _75.88_ | _52.03_ | _54.42_ | 67.39 | 57.48 | _75.76_ | _77.56_ | 77.46 | 69.92 |
| | _Ours_ | **65.60** | **64.71** | _88.79_ | **76.61** | **46.80** | **43.62** | **69.66** | **61.92** | **74.18** | **73.31** | _83.5_ | _74.54_ |
| VGG | Original | 95.77 | 94.15 | 95.73 | 85.56 | 96.26 | 95.07 | 94.57 | 87.16 | 97.67 | 98.06 | 97.14 | 90.67 |
| | Retrain | 52.41 | 50.35 | 96.13 | 78.14 | 45.40 | 41.49 | 78.84 | 67.81 | 77.47 | 75.89 | 97.03 | 86.25 |
| | SISA | 66.91 | 78.20 | **86.34** | **65.13** | 61.87 | 62.41 | 67.02 | _63.18_ | 82.31 | 83.30 | 85.04 | 70.14 |
| | INF | 73.53 | 75.67 | 77.28 | 62.79 | 75.33 | 76.08 | **76.36** | **65.69** | 83.11 | 85.63 | _86.08_ | _71.16_ |
| | IUPC | _58.65_ | _61.57_ | 73.33 | 60.09 | _55.18_ | _58.37_ | 61.04 | 56.97 | _80.07_ | _78.12_ | 70.95 | 56.97 |
| | _Ours_ | **53.84** | **51.74** | _78.61_ | _64.60_ | **48.37** | **46.46** | _71.34_ | 59.87 | **78.68** | **76.59** | **87.49** | **72.05** |
| ViT | Original | 95.65 | 94.71 | 96.34 | 85.47 | 84.93 | 84.96 | 78.70 | 69.90 | 97.67 | 98.06 | 98.44 | 92.51 |
| | Retrain | 68.03 | 66.09 | 94.01 | 78.64 | 28.17 | 25.28 | 72.39 | 64.10 | 73.02 | 71.89 | 95.11 | 87.24 |
| | SISA | _76.57_ | 79.61 | 87.67 | 74.15 | _46.79_ | _47.24_ | 67.25 | 58.68 | 82.96 | 84.37 | _93.73_ | 82.30 |
| | INF | 81.72 | 83.67 | **91.88** | **76.54** | 54.99 | 56.28 | _68.06_ | _60.95_ | 81.74 | 83.01 | **94.61** | **85.35** |
| | IUPC | 76.88 | _77.93_ | 88.25 | 74.71 | 50.85 | 53.67 | 67.33 | 59.64 | _76.85_ | _78.92_ | 89.07 | 81.25 |
| | _Ours_ | **71.07** | **70.11** | _89.98_ | _75.43_ | **31.59** | **29.86** | **69.45** | **61.24** | **74.77** | **71.36** | 91.18 | _83.52_ |

current $T$-th round $Acc_F$. ii) **Life-long forgetting**: Average accuracy on the forget set of past $T-1$ rounds $Acc_P$. iii) **Model utility**: Accuracy on the retain set of current $T$-th round $Acc_R$. iv) **Generalization**: Accuracy on the testing set $Acc_T$. v) **Time efficiency**: Average running time of one unlearning round.

Please refer to Appendix E.2 for more details of experimental settings.

## 3.2 MAIN RESULTS

**Results on Various Datasets.** We include 5 consecutive rounds of unlearning requests to simulate real-world scenarios. We report the average performance of 5 rounds in Table 1. Due to the space limit, detailed results of each round are provided in Appendix F. As shown in Table 1, our proposed method significantly outperforms compared baselines in terms of overall performance after handling sequential unlearning requests. Specifically, i) *our method ensures life-long forgetting throughout all unlearning rounds while maintaining model performance on retain set ($Acc_R$) and testing set ($Acc_T$).* We observe that some classic unlearning methods, such as SISA and INF, perform poorly in life-long forgetting, which is reflected by $Acc_P$ on the past forget set. After consecutive unlearning rounds, the performance on the $Acc_P$ metric of these methods is even higher than the $Acc_F$ metric of previous rounds, indicating a significant flaw in life-long forgetting. This is because, although these methods do not explicitly utilize the retain set, the knowledge of the retain set still resides in the sub-models of SISA and the Hessian matrix of INF. However, there are samples in the retain set that are similar to those in past forget sets. This similarity causes the decision boundaries of the unlearned model to lack clear distinctions between previously forgotten samples and similar samples in the retain set, thereby hindering effective life-long forgetting. Our method maintains a lower value on $Acc_P$ because it considers past forget sets in design, thus dynamically adjusting the decision boundaries of the unlearned model. In other words, the model's decision boundaries post-unlearning create a separation between the current retain set and past forget sets, even if they are similar. *On $Acc_F$, we improve by 9.05% compared to the best baseline (IUPC), and on $Acc_R$, we are only 8.00% behind retraining, indicating that our unlearning method is effective. On $Acc_T$, we are 3.1% lower than the best baseline (INF).* However, the high generalization performance of INF comes at the cost of sacrificing some degree of unlearning. INF does not completely erase each data point, but instead weakens the model's overfitting to specific data points through local weight adjustments. This makes the model's dependence on the learned data smoother, thereby preserving its generalization ability. For this reason, INF only reduces the influence of certain data points rather than completely removing them, and its unlearning effect is relatively limited. ii) *Experiments on larger datasets and models in Appendix F.1 and F.2.* Additionally, the experimental results for special unlearning subset selection, such as unlearning an entire class in each round, can be found in Appendix F.6.

**Different Numbers of Unlearning Rounds.** We further investigate different numbers of unlearning rounds. Prior work (Wichert & Sikdar, 2024; Georgiev et al., 2024; Koloskova et al., 2025) has shown that when the retain set is too small (i.e., a larger portion of data is unlearned), model performance degrades significantly. In such cases, retraining the model becomes a more viable option,

as the computational cost is no longer prohibitive. Accordingly, we impose a limit on the number of samples to be unlearned, ensuring that after multiple unlearning rounds, the remaining dataset size remains within a reasonable range. Then, we adjust the number of unlearning rounds to 5, 10, 15, and 20. We conduct experiments using ResNet-18 on CIFAR-10 and report the results in Table 2. The average results (with detailed data in Appendix F) indicate that our method, when handling a larger number of unlearning rounds, outperforms the existing compared baselines across various performance metrics. These results sufficiently demonstrate the effectiveness and stability of our method in sequential unlearning. Furthermore, we observe that as unlearning

| Rounds | Method | $Acc_F(\downarrow)$ | $Acc_P(\downarrow)$ | $Acc_R(\uparrow)$ | $Acc_T(\uparrow)$ |
|---|---|---|---|---|---|
| $T = 5$ | Retrain | 60.26 | 55.21 | 94.17 | 81.15 |
| | SISA | 76.46 | 78.54 | 93.85 | **81.13** |
| | INF | 87.93 | 87.97 | **95.25** | 75.37 |
| | IUPC | 70.73 | 72.82 | 84.28 | 74.95 |
| | *Ours* | **63.39** | **61.43** | 88.78 | 76.61 |
| $T = 10$ | Retrain | 56.30 | 53.87 | 90.43 | 78.40 |
| | SISA | 72.15 | 73.93 | 91.03 | **76.98** |
| | INF | 85.25 | 86.21 | **93.03** | 73.37 |
| | IUPC | 67.86 | 69.41 | 79.11 | 70.09 |
| | *Ours* | **57.59** | **55.71** | 84.05 | 74.75 |
| $T = 15$ | Retrain | 50.11 | 49.93 | 87.75 | 76.60 |
| | SISA | 69.93 | 70.57 | 85.68 | 70.67 |
| | INF | 82.19 | 85.79 | **88.30** | 70.12 |
| | IUPC | 62.53 | 65.74 | 73.17 | 65.31 |
| | *Ours* | **53.27** | **50.38** | 81.84 | **71.01** |
| $T = 20$ | Retrain | 47.30 | 45.93 | 82.25 | 70.87 |
| | SISA | 61.15 | 63.97 | 76.61 | 65.84 |
| | INF | 77.52 | 79.73 | **83.49** | 67.87 |
| | IUPC | 57.51 | 63.36 | 67.95 | 60.18 |
| | *Ours* | **50.78** | **48.24** | 78.19 | **70.53** |

Table 2: Effect of different numbers of unlearning rounds. Except for Retrain, the best results are highlighted in **bold**.

rounds increase, some traditional methods such as SISA and IUPC exhibit significant fluctuations in model performance. This is primarily because these methods conduct single-round unlearning, and multiple rounds lead to instability in model performance. In contrast, our method reduces instability by fine-tuning the original model on forget set in each unlearning round and establishing connections between them to obtain the optimal model, thereby ensuring a smooth transition in performance.

**Time Efficiency.** When studying the issue of sequential unlearning, assessing the efficiency of the method is extremely crucial. We evaluate efficiency by the average running time of each round. Specifically, we conduct experiments using ResNet-18 on CIFAR10 when handling 5, 10, 15, and 20 unlearning rounds. It is evident from Figure 3 that IUPC exhibits a significant disadvantage in terms of unlearning efficiency. This is because IUPC needs to generate adversarial samples in each round, and the running time of this process is greatly influenced by the number of adversarial samples generated. Since increasing unlearning rounds reduces the size of the forget set in each round, thereby reducing the

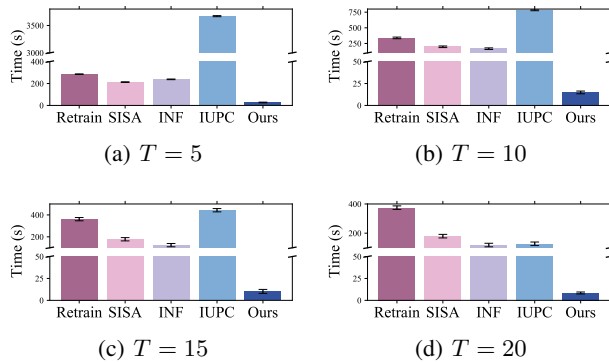

(a) $T = 5$ (b) $T = 10$

(c) $T = 15$ (d) $T = 20$

Figure 3: Average running time of one unlearning round.

number of adversarial samples, the efficiency of IUPC is enhanced with larger unlearning rounds. Although SISA shows some improvement over retraining, its time overhead remains substantial. This is mainly because unlearning requests from multiple rounds are scattered in multiple shards, which increases the retraining overhead of SISA. INF needs to compute the Hessian matrix of the current unlearning dataset relative to the previous round's optimal model at each unlearning round, which is time-consuming. Our method, because it pre-computes and stores the Hessian matrix during initialization, only requires fine-tuning the original model on the forget set, greatly reducing the computational load. This makes our method significantly more efficient in sequential unlearning. Additionally, the sensitivity of our method's unlearning efficiency to dataset size and model scale is relatively low, making it more broadly applicable in practical settings.

**Membership Inference Attack.** Following previous research, we validate the unlearning effectiveness of our proposed method using Membership Inference Attack (MIA). We report the detailed settings of MIA in Appendix E.3. As observed from Table 3, under the MIA evaluation, the retraining method exhibits the best unlearning performance, approaching 0.

Our proposed method significantly outperforms the baseline methods, nearing the performance of retraining, indicating that our approach possesses sufficient unlearning effectiveness. Additionally, it is observed that the SISA method demonstrates limited unlearning effectiveness in this sequential unlearning setting, which we attribute to the ensemble characteristics of SISA that confer it with stronger generalization capabilities; even if one sub-model is retrained, the remaining ensemble retains some generalization ability.

| Model | Method | CIFAR-10 | CIFAR-100 | Fashion-MNIST |
|-------|--------|----------|-----------|---------------|
| ResNet-18 | Original | 0.8608 | 0.9168 | 0.7858 |
| | Retrain | 0.0005 | 0.0154 | 0.0019 |
| | SISA | 0.6876 | 0.6268 | 0.7419 |
| | INF | 0.2328 | 0.2096 | 0.0667 |
| | IUPC | _0.0749_ | _0.1042_ | _0.0319_ |
| | ***Ours*** | **0.0245** | **0.0569** | **0.0057** |
| VGG | Original | 0.8226 | 0.8228 | 0.7764 |
| | Retrain | 0.0106 | 0.0164 | 0.0224 |
| | SISA | 0.5238 | 0.7001 | 0.5342 |
| | INF | 0.1302 | 0.1494 | 0.0540 |
| | IUPC | _0.0462_ | **0.0256** | _0.0176_ |
| | ***Ours*** | **0.0215** | _0.0436_ | **0.0064** |

Table 3: MIA on various datasets. Except for Original and Retrain, the best results are highlighted in **bold**, and secondary results are highlighted with _underline_.

### 3.3 MORE RESULTS

**Pseudo Label Selection.** The method for selecting pseudo-labels is not fixed. A natural choice is to maximize confusion by assigning pseudo-labels based on the closest misclassified label at the start of training, which we refer to as *Misclassified*. The motivation is to minimize the performance degradation of the model, as this setup helps to reduce changes in the model's decision boundary to the greatest extent. In addition, following the setup in prior work (Golatkar et al., 2021; Cha et al., 2024; Li et al., 2025), another popular method is to define the pseudo-label as a uniformly weighted vector that matches the length of the model's softmax output layer. Specifically, each element of the pseudo-label is set to $1/n$, where $n$ is the length of the vector; we refer to this method as *Uniform*. We compare these strategies on ResNet-18 and VGG, with results presented in Table 4. It can be observed that in most cases, the Uniform setting leads to better unlearning performance. Although this stricter approach may reduce model generalization compared to Misclassified, the trade-off remains acceptable. Therefore, we adopt the Uniform strategy throughout this work.

Table 4: Choice of pseudo-labels. Except for Retrain, the best results are highlighted in **bold**.

| Model | Method | CIFAR-10 | | | | CIFAR-100 | | | | Fashion-MNIST | | | |
|-------|--------|----------|----------|----------|----------|-----------|----------|----------|----------|---------------|----------|----------|----------|
| | | $Acc_F(\downarrow)$ | $Acc_P(\downarrow)$ | $Acc_R(\uparrow)$ | $Acc_T(\uparrow)$ | $Acc_F(\downarrow)$ | $Acc_P(\downarrow)$ | $Acc_R(\uparrow)$ | $Acc_T(\uparrow)$ | $Acc_F(\downarrow)$ | $Acc_P(\downarrow)$ | $Acc_R(\uparrow)$ | $Acc_T(\uparrow)$ |
| ResNet-18 | Original | 95.77 | 95.15 | 96.18 | 90.56 | 96.26 | 95.07 | 96.02 | 88.16 | 97.67 | 97.06 | 98.44 | 90.67 |
| | Retrain | 62.12 | 58.27 | 95.76 | 82.56 | 31.60 | 27.45 | 70.97 | 63.87 | 73.81 | 66.15 | 94.68 | 83.17 |
| | *Misclassified* | 72.50 | 70.23 | 88.19 | **77.96** | 52.73 | 51.33 | 67.72 | **63.35** | 79.60 | 78.90 | **85.84** | **75.79** |
| | *Uniform* | **65.60** | **64.71** | **88.79** | 76.61 | **46.80** | **43.62** | **69.66** | 61.92 | **74.18** | **73.31** | 83.50 | 74.54 |
| VGG | Original | 95.77 | 94.15 | 95.73 | 85.56 | 96.26 | 95.07 | 94.57 | 87.16 | 97.67 | 98.06 | 97.14 | 90.67 |
| | Retrain | 52.41 | 50.35 | 96.13 | 78.14 | 45.40 | 41.49 | 78.84 | 67.81 | 77.47 | 75.89 | 97.03 | 86.25 |
| | *Misclassified* | 58.03 | 55.61 | 77.21 | 63.82 | 50.08 | 47.17 | **72.85** | **61.47** | 82.00 | 79.31 | 87.32 | **72.74** |
| | *Uniform* | **53.84** | **51.74** | **78.61** | **64.60** | **48.37** | **46.46** | 71.34 | 59.87 | **78.68** | **76.59** | **87.49** | 72.05 |

**Ablation Study.** Table 5 presents the results of ablation studies conducted using ResNet-18 on CIFAR10. We sequentially remove the 1st term loss $\sum_{t=1}^{T-1} \ell_F^t(\boldsymbol{\theta})$ on the past forget sets and the 3rd term loss

| 1st Term | 2nd Term | 3rd Term | $Acc_F(\downarrow)$ | $Acc_P(\downarrow)$ | $Acc_R(\uparrow)$ | $Acc_T(\uparrow)$ |
|----------|----------|----------|---------------------|---------------------|-------------------|-------------------|
| ✓ | ✓ | ✓ | 65.60 | 64.71 | 88.79 | 76.61 |
| ✗ | ✓ | ✓ | 58.57 | 66.28 | 71.00 | 69.51 |
| ✓ | ✓ | ✗ | 39.18 | 37.61 | 56.64 | 53.96 |
| ✗ | ✓ | ✗ | 0.08 | 40.78 | 50.17 | 48.68 |

Table 5: Results of ablation studies on each term in Eq. (5).

$\ell_R^T(\boldsymbol{\theta})$ on the current retain set to assess their impact on the overall unlearning performance. Initially, when we remove the loss on the past forget sets (1st term), a significant increase in the life-long forgetting performance metric $Acc_P$ is observed. This indicates that without proper handling of earlier forgotten data, the model regains its memory of earlier forgotten data, resulting in reduced life-long forgetting performance. Subsequently, when we remove the loss on the current retain set (3rd term), although the model still performs well on metrics $Acc_F$ and $Acc_P$, there is a noticeable decline in performance on metrics $Acc_R$ and $Acc_T$. This suggests that removing this component of loss, while allowing the model to handle unlearning requests, significantly compromises its adaptability to testing data and performance on non-forgotten data. These results from the ablation studies clearly demonstrate that the loss on the past forget sets and the loss on the current retain set play crucial roles in our sequential unlearning method. The former primarily affects the effectiveness of life-long forgetting, while the latter ensures that the model maintains good performance and generalization ability when processing new unlearning requests.

**Hyperparameter.** We use ResNet-18 to explore the effect of hyperparameter $\alpha$ on unlearning performance ($Acc_F$ and $Acc_P$), utility ($Acc_R$), and generalization ability ($Acc_T$). As shown in Figure 4, when the value of $\alpha$ is low, the model exhibits reduced utility and generalization ability after unlearning. As the value of $\alpha$ increases, both the utility and generalization ability of the model gradually improve, approaching the

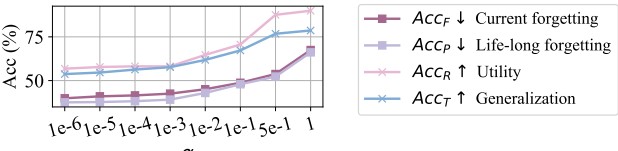

Figure 4: Effect of hyperparameter $\alpha$ w.r.t. effectiveness of unlearning on CIFAR10. We report more results on other datasets in Appendix F.

levels achieved through retraining. However, the unlearning performance of the model decreases with an increase in $\alpha$. At $\alpha = 0.5$, the experimental results show that the model achieves an ideal balance between unlearning performance and utility/generalization ability. This suggests that $\alpha = 0.5$ is an appropriate setting that ensures both effective unlearning of the forget set and reliable utility performance on the retain set.

**Qualitative Analysis.** To conduct a deeper analysis, we examine the differences in parameters of unlearned models between baseline methods and those obtained through retraining. Figure 6 in Appendix F displays a heatmap of the CKA correlations between the unlearned model and the retrained model. The results indicate that our method is most closely aligned with retraining from scratch at the parameter level. This insight is consistent with the theoretically optimal results described in Section C, where the theoretically optimal model denotes the one obtained through retraining.

**Sequential Unlearning vs. Batch Unlearning.** To further clarify the differences between sequential and batch unlearning, we conduct a validation experiment. Specifically, we first evaluate the performance of our proposed method after $T$ rounds of unlearning, then assume that all unlearning requests arrive simultaneously, i.e., setting $T = 1$, to compare the results.

| Rounds | Method | $Acc_F(\downarrow)$ | $Acc_R(\uparrow)$ | $Acc_T(\uparrow)$ |
|---|---|---|---|---|
| CIFAR-10 | Batch Unlearning | 44.64 | 55.72 | 48.16 |
| | Sequential Unlearning | 41.60 | 71.04 | 61.48 |
| CIFAR-100 | Batch Unlearning | 57.70 | 65.51 | 49.59 |
| | Sequential Unlearning | 40.60 | 67.88 | 51.31 |
| Fashion-MNIST | Batch Unlearning | 64.16 | 79.77 | 69.52 |
| | Sequential Unlearning | 54.67 | 95.83 | 86.39 |

Table 6: Comparison of sequential unlearning and batch unlearning with ResNet-18 on the same unlearning task.

The experimental results are reported in Table 6. The results indicate that, compared to batch unlearning, sequential unlearning generally yields a lower $Acc_F$, suggesting that sequential unlearning is superior in terms of unlearning efficacy. We hypothesize that this may be due to the larger volume of unlearning samples processed during batch unlearning, which can easily conflict with preserving model performance, leading to suboptimal unlearning outcomes. Meanwhile, batch unlearning methods process a large number of samples simultaneously, which can make the unlearning step difficult to control, often resulting in over-unlearning, instability, and significant performance degradation. In contrast, sequential unlearning proceeds incrementally, removing fewer samples per round and allowing the model to adapt gradually. This controlled approach helps mitigate over-unlearning and promotes more stable outcomes.

## 4  CONCLUSION

In this paper, we focus on sequential unlearning, a practical problem that MU faces in real-world scenarios. In these situations, unlearning requests are received sequentially and must be processed instantly upon arrival. Existing unlearning methods cannot be directly applied to sequential unlearning due to two main challenges, i.e., the incapacity for lifelong forgetting and inefficiency in handling sequential unlearning requests. To address these challenges, we follow the idea of continual learning and propose a novel method tailored for sequential unlearning. Firstly, we introduce a lifelong forgetting term into the unlearning objective to ensure that the unlearned data will not be re-memorized in subsequent rounds. Secondly, we establish a source-free optimization to eliminate the need for the retain set and past forget sets, thereby greatly enhancing efficiency. We conducted extensive experiments to demonstrate the effectiveness and efficiency of our proposed method. Although MU has gained popularity and has been extensively studied in recent years, it has not been widely applied in practice due to unresolved practical issues. This paper addresses the issue of the instant processing requirement of sequentially received requests, bridging the gap in practical use, and offering valuable insights for the application of MU.

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

## A    RELATED WORK

Based on the level of unlearning completeness, i.e., the quality of forgetting, unlearning methods can be categorized into two main approaches: exact unlearning and approximate unlearning.

### A.1    EXACT UNLEARNING

This approach aims to achieve full completeness by efficient retraining, i.e., to mimic retraining from scratch using the updated dataset excluding the unlearned data points. SISA pioneers exact unlearning through a shard-aggregation approach, i.e., dividing the data into shards, retraining each shard, and aggregating the sub-models (Bourtoule et al., 2021). However, due to the nature of retraining, exact unlearning encounters significant computational challenges in sequential unlearning.

### A.2    APPROXIMATE UNLEARNING

This approach aims to approximate the parameters or the output of the ground truth model, i.e., retraining from scratch using the updated dataset that excludes the unlearning target. There are two distinct lines of research, i.e., reverse unlearning and active unlearning. **Reverse Unlearning** estimates the influence of target data and directly removes it through reverse gradient operations (Sekhari et al., 2021; Wu et al., 2022; Mehta et al., 2022; Jang et al., 2023). However, accurately estimating influence remains an ongoing challenge for deep learning models (Basu et al., 2021; Guo et al., 2021). Inaccurate estimation can accumulate and become even more problematic in sequential unlearning. **Active Unlearning** follows the idea of continual learning (i.e., learning to unlearn), continuously training the model to overwrite the memory of target data (Ma et al., 2022; Li et al., 2023b; Cha et al., 2024). However, similar to the issue of catastrophic forgetting faced in continual learning, active unlearning cannot meet the lifelong forgetting requirements in the context of sequential unlearning (as demonstrated by our empirical study in Figure 1).

### A.3    SEQUENTIAL UNLEARNING

Sequential unlearning, also known as stream removal, is a challenging problem that existing unlearning methods cannot adequately address (Nguyen et al., 2022). As mentioned above, existing methods face challenges such as the incapability of life-long forgetting, inaccurate estimation, and inefficiency. Gupta et al. (2021) studies the adaptive MU problem, where requests are sequential and interdependent (i.e., adaptive). This paper focuses on the non-adaptive setting, where there is no prior knowledge about unlearning requests and no interdependence among them.

## B    MORE DISCUSSION

### B.1    FURTHER EXPLANATION OF MOTIVATION

We conduct a further analysis of the experimental results presented in Figure 1. Specifically, we focus on the underperformance of SISA, as one would reasonably expect its modular nature to offer advantages in sequential unlearning scenarios. However, its actual performance in Figure 1 is suboptimal. We attribute this to the following reasons. Although SISA is defined as exact unlearning at the algorithmic level, its practical performance often falls short of expectations. This is primarily due to its distributed aggregation design: during each forgetting operation, only one sub-model is updated, while the remaining sub-models may still retain the relevant knowledge. As a result, there is a gap between its unlearning performance and the ideal state. This phenomenon has been highlighted in previous studies (Wu et al., 2022; Chen et al., 2024). In sequential unlearning scenarios, such deviations progressively accumulate with each round of unlearning operations, thereby exacerbating the severity of the problem.

### B.2    DISCUSSION ON THE ASSUMPTIONS' RATIONALITY AND ERROR ACCUMULATION

The errors introduced by our method primarily arise from a set of well-established assumptions, i.e., the Taylor expansion (Koh & Liang, 2017; Mirzadeh et al., 2020), the vanishing-gradient assumption (He et al., 2015; Koh et al., 2019), and the Hessian approximation (Ghorbani et al., 2019). These

assumptions have been extensively validated in prior work and are likewise applicable in the experimental settings employed in our study. As demonstrated in Table 11, our empirical results show that error does not accumulate appreciably across successive unlearning rounds, and the framework's performance remains stable through all rounds.

### B.3 DISCUSSION ON APPLICATION SCENARIOS

Since existing research on unlearning in classification models mainly focuses on visual tasks, we have simply followed the setup from previous work in our experiments. However, our framework does not assume a specific model structure and can be easily extended to other classification tasks, such as text classification, graph node classification, or subgraph classification. In these tasks, the setup for pseudo-labels is the same as in visual tasks. Additionally, unlearning for generative models is also theoretically feasible, as our method essentially addresses an optimization problem that can be adapted to the unlearning scenario in generative models. The main difference between the two lies in the setup of pseudo-labels. For example, in image generation models (Li et al., 2024), pseudo-labels could be set as Gaussian noise; in text generation models (Wang et al., 2024; Ji et al., 2024), pseudo-labels could be set as a uniform distribution or involve rejecting certain content (such as "I don't know").

## C DERIVATION OF EQ. EQUATION 11 VIA INFLUENCE FUNCTIONS

In this appendix, we provide the detailed derivation of Eq. equation 11, which approximates the difference between the model parameters trained on successive retain sets, using classical influence function theory (Koh & Liang, 2017; Koh et al., 2019).

We aim to estimate the following term: $\hat{\boldsymbol{\theta}}_R^T - \hat{\boldsymbol{\theta}}_R^{T-1}$. According to the decomposition already introduced in the main text:

$$\hat{\boldsymbol{\theta}}_R^T - \hat{\boldsymbol{\theta}}_R^{T-1} \approx (\hat{\boldsymbol{\theta}}_R^T - \boldsymbol{\theta}^0) - (\hat{\boldsymbol{\theta}}_R^{T-1} - \boldsymbol{\theta}^0).$$

Since $R_0 \setminus R_T = \bigcup_{t=1}^T F_t$, i.e., the set of all unlearned samples up to round $T$, we apply the influence function approximation:

$$\hat{\boldsymbol{\theta}}_R^T - \boldsymbol{\theta}^0 \approx -\frac{|R_0 \setminus R_T|}{|R_0|} H_{\boldsymbol{\theta}^0}^{-1} \nabla g_R^T(\boldsymbol{\theta}^0),$$

where we define $\nabla g_R^T(\boldsymbol{\theta}^0) = \sum_{(x,y) \in R_0 \setminus R_T} \nabla \ell(f_{\boldsymbol{\theta}^0}(x), y)$.

Similarly, we have:

$$\hat{\boldsymbol{\theta}}_R^{T-1} - \boldsymbol{\theta}^0 \approx -\frac{|\bigcup_{t=1}^{T-1} F_t|}{|R_0|} H_{\boldsymbol{\theta}^0}^{-1} \nabla \ell_F^{T-1}(\boldsymbol{\theta}^0).$$

Subtracting these two terms, we can get

$$\hat{\boldsymbol{\theta}}_R^T - \hat{\boldsymbol{\theta}}_R^{T-1} = \left[-\frac{|\bigcup_{t=1}^T F_t|}{|R_0|} H_{\boldsymbol{\theta}^0}^{-1} \nabla \ell_F^T\right] - \left[-\frac{|\bigcup_{t=1}^{T-1} F_t|}{|R_0|} H_{\boldsymbol{\theta}^0}^{-1} \nabla \ell_F^{T-1}\right].$$

Note that:

$$\bigcup_{t=1}^T F_t = \left(\bigcup_{t=1}^{T-1} F_t\right) \cup F_T.$$

Therefore, we have:

$$\hat{\boldsymbol{\theta}}_R^T - \hat{\boldsymbol{\theta}}_R^{T-1} \approx \frac{|\bigcup_{t=1}^{T-1} F_t|}{|R_0|} H_{\boldsymbol{\theta}^0}^{-1} \nabla \ell_F^{T-1}(\boldsymbol{\theta}^0) + \frac{|F_T|}{|R_0|} H_{\boldsymbol{\theta}^0}^{-1} \nabla \ell_F^T(\boldsymbol{\theta}^0).$$

## D ALGORITHM PROCEDURE AND COMPUTATIONAL IMPLEMENTATION

### D.1 ALGORITHM PROCEDURE.

We summarize our proposed framework in Algorithm 1.

---

**Algorithm 1** Sequential unlearning algorithm

---

**Input**: Original model $f_{\boldsymbol{\theta}^0}$, initial retain set $R_0$, the number of unlearning rounds $T$, the adjustable parameter $\alpha$, learning rate $\eta$. **Procedure**:

1: Initialize $t = 0$, $f_{\boldsymbol{\theta}_*^t} = f_{\boldsymbol{\theta}^0}$, $\nabla g_R^0(\boldsymbol{\theta}^0) = 0$
2: Compute and save Hessian: $H_{\boldsymbol{\theta}^0}$
3: Let $t = 1$
4: **while** $t \leq T$ **do**
5:     Receive the unlearning request $F_t$
6:     Create a copy $f_{\hat{\boldsymbol{\theta}}_F^t}$ of the original model
7:     **while** Not Converged **do**
8:         Compute loss: $\ell_F^T(\hat{\boldsymbol{\theta}}_F^t) = \sum_{(x,y) \in F_t} \ell(f_{\hat{\boldsymbol{\theta}}_F^t}(x), \hat{y})$
9:         Update parameter: $\hat{\boldsymbol{\theta}}_F^t = \hat{\boldsymbol{\theta}}_F^t - \eta \nabla \ell_F^t(\hat{\boldsymbol{\theta}}_F^t)$
10:     **end while**
11:     Compute gradient: $\nabla \ell_F^t(\boldsymbol{\theta}^0)$
12:     Update parameter: $\boldsymbol{\theta}_*^t$ according to Eq. (12)
13:     Update: $\nabla g_R^t(\boldsymbol{\theta}^0) = \nabla g_R^{t-1}(\boldsymbol{\theta}^0) + \nabla \ell_F^t(\boldsymbol{\theta}^0)$
14: **end while**
15: **return** The unlearned model $f_{\boldsymbol{\theta}_*^T}$

---

## D.2 COMPUTATIONAL IMPLEMENTATION (HESSIAN MATRIX)

In our framework, the Hessian only needs to be computed once before conducting unlearning. For models with a smaller parameter scale, we use a diagonal approximation (Elsayed et al., 2024) to compute the Hessian matrix, rather than calculating it exactly. We validate the approximation on different models. To measure the error between the approximated Hessian matrix and the true Hessian matrix, we used the Frobenius norm to compute the square root of the sum of squared differences between the corresponding matrix elements. The experimental results are shown in Table 7. As can be seen, traditional convolutional neural networks with simpler structures, such as ResNet-18 and VGG, have relatively small errors. Models with higher complexity, such as ViT and Swin Transformer, exhibit larger errors in their diagonal approximations, but the errors are still within a reasonable range.

Table 7: Hessian matrix computation errors on different datasets and models.

| Model | Dataset | | | |
|---|---|---|---|---|
| | CIFAR-10 | CIFAR-100 | Fashion-MNIST | ImageNet-1K |
| ResNet-18 | $1.26 \times 10^{-4} \sim 3.07 \times 10^{-4}$ | $2.31 \times 10^{-4} \sim 4.15 \times 10^{-4}$ | $1.18 \times 10^{-4} \sim 1.79 \times 10^{-4}$ | $3.06 \times 10^{-4} \sim 6.44 \times 10^{-4}$ |
| VGG | $2.68 \times 10^{-4} \sim 5.31 \times 10^{-4}$ | $3.57 \times 10^{-4} \sim 6.11 \times 10^{-4}$ | $1.86 \times 10^{-4} \sim 2.77 \times 10^{-4}$ | $4.83 \times 10^{-4} \sim 8.27 \times 10^{-4}$ |
| VIT | $5.18 \times 10^{-4} \sim 9.42 \times 10^{-4}$ | $6.37 \times 10^{-4} \sim 12.53 \times 10^{-4}$ | $4.21 \times 10^{-4} \sim 7.39 \times 10^{-4}$ | $6.48 \times 10^{-4} \sim 13.91 \times 10^{-4}$ |
| Swin Transormer | $6.65 \times 10^{-4} \sim 12.73 \times 10^{-4}$ | $7.82 \times 10^{-4} \sim 13.07 \times 10^{-4}$ | $5.57 \times 10^{-4} \sim 8.05 \times 10^{-4}$ | $8.09 \times 10^{-4} \sim 15.37 \times 10^{-4}$ |

For ultra-large-scale models, the computational cost of the diagonal approximation becomes substantial. Therefore, we employ the Hessian-Free assumption to substitute the full Hessian with a scaled identity matrix, which is both simple and effective in large-scale classification models as demonstrated by (Zhang et al., 2022) and (Fan et al., 2023), and in LLM settings as shown by (Jia et al., 2024). Empirical results indicate that this approximation achieves strong performance with minimal computational overhead.

# E EXPERIMENTAL SETTINGS

## E.1 BASELINE IMPLEMENTATION DETAILS

In our experiments, the implementation details for each baseline method compared are as follows:

- **Retrain**: Upon receiving an unlearning request, the model is directly retrained on the retain set corresponding to the current round until optimal performance is achieved on the retain set.

- **SISA** (Bourtoule et al., 2021): The dataset is divided into multiple subsets, with each subset training an independent sub-model. The final model output is an aggregation of these sub-model outputs. When specific data needs to be unlearned, only the sub-model containing this data is retrained. In this paper, we opt to divide the dataset into 10 subsets, with each subset independently training a sub-model.

- **INF** (Sekhari et al., 2021): This method is based on influence functions, directly calculating the impact of removing a specific forget set on the parameters of the previous round's optimal model, and updating the model parameters based on this impact. Constrained by GPU memory constraints, we employ the Hessian-vector product (HVP) technique for computing the Hessian matrix.

- **IUPC** (Cha et al., 2024): This method employs unlearning through adversarial samples. Initially, adversarial samples are generated using the unlearning set and PGD (Madry et al., 2017) attack, which assist in maintaining the original model's decision boundary. The unlearning process includes three loss terms, i.e. classification loss on the forget set with mixed classes, classification loss on adversarial samples, and a parameter regularization term. The parameter regularization term is formed by calculating the importance of parameters for predictions on the forget set with MAS (Aljundi et al., 2018) and the squared difference between the updated model parameters and the original model parameters. In implementing the IUPC baseline, we employ L2-PGD (Ilyas et al., 2019) attacks to generate adversarial examples, with specific parameters including a learning rate of $1e-1$, attack iterations of 100, and perturbation bound $\epsilon$ set at 0.4, with each forget set sample correspondingly generating 20 adversarial examples.

### E.2 EXPERIMENTAL DETAILS

In our experiments, we segment the sequential unlearning process into $T$ rounds to simulate the continuous unlearning demands that arise in real-world applications. Specifically, we unlearn 5000 samples in total. The number of samples in each unlearning is determined by the number of total rounds. During each unlearning round, we randomly sample the corresponding number of samples from the previous round's retain set to ensure the randomness and practicality of the unlearning process. For unlearning, we use an SGD optimizer with a learning rate of $1e-3$, weight decay of $1e-5$, and momentum of 0.9 across all experiments. These parameters are chosen based on best practices observed in preliminary experiments. All experiments can be conducted on a single Nvidia 4090 GPU.

### E.3 MIA DETAILS

Based on previous research (Choi & Na, 2023), we employ a Membership Inference Attack (MIA) to evaluate the unlearning performance of MU algorithms. During the evaluation phase, we train a binary classifier $\psi(\cdot)$ to simulate the attacker's behavior, aiming to distinguish between the loss values of the unlearned data $\boldsymbol{x}_f$ and unseen data $\boldsymbol{x}_u$ (i.e., not in the training set). The attacker aims to determine whether data $\boldsymbol{x}_f$ were used during training. Ideally, the performance of this binary classifier $\psi(\cdot)$ should be as follows: $\psi(x) = 1$ if $x \in \boldsymbol{x}_f$ and $\psi(x) = 0$ if $x \in \boldsymbol{x}_u$. If the prediction accuracy of $\psi(\cdot)$ is 0.5, it indicates that the MU algorithm performs well, suggesting that $\psi(\cdot)$ is unable to distinguish between unlearned sample $\boldsymbol{x}_f$ and unseen samples $\boldsymbol{x}_u$. Furthermore, we denote $M$ as the accuracy of $\psi(\cdot)$ and define the unlearning score as $2 \times |M - 0.5|$ to assess the unlearning effectiveness on data $\boldsymbol{x}_f$, where a lower score is preferable.

## F EMPIRICAL RESULTS

### F.1 RESULTS ON VARIOUS DATASETS

We present the detailed unlearning performance of different models on three real-world datasets. The number of unlearning rounds is set to 5. The experimental results using ResNet-18, VGG, and ViT are reported in Tables 8, 9, and 10, respectively.

Table 8: The relationship between the results on different datasets and the effects of sequential unlearning when the model is ResNet-18.

| Method | Round ($T$) | CIFAR-10 | | | | CIFAR-100 | | | | Fashion-MNIST | | | |
|---|---|---|---|---|---|---|---|---|---|---|---|---|---|
| | | $Acc_F(\downarrow)$ | $Acc_P(\downarrow)$ | $Acc_R(\uparrow)$ | $Acc_T(\uparrow)$ | $Acc_F(\downarrow)$ | $Acc_P(\downarrow)$ | $Acc_R(\uparrow)$ | $Acc_T(\uparrow)$ | $Acc_F(\downarrow)$ | $Acc_P(\downarrow)$ | $Acc_R(\uparrow)$ | $Acc_T(\uparrow)$ |
| Original | - | 95.77 | 95.15 | 96.18 | 90.56 | 96.26 | 95.07 | 96.02 | 88.16 | 97.67 | 97.06 | 98.44 | 90.67 |
| Retrain | 1 | 65.56 | - | 98.62 | 85.17 | 36.38 | - | 91.63 | 79.47 | 78.75 | - | 97.50 | 85.95 |
| | 2 | 64.30 | 58.38 | 96.33 | 84.77 | 32.94 | 65.65 | 89.36 | 77.24 | 74.18 | 67.79 | 95.01 | 83.66 |
| | 3 | 63.57 | 58.20 | 95.64 | 82.42 | 31.82 | 64.07 | 88.91 | 76.19 | 73.87 | 65.52 | 94.88 | 82.89 |
| | 4 | 59.98 | 57.68 | 94.55 | 81.20 | 30.30 | 64.06 | 88.16 | 75.99 | 71.92 | 64.71 | 94.31 | 82.77 |
| | 5 | 57.19 | 57.19 | 93.65 | 79.25 | 26.55 | 61.47 | 85.88 | 74.17 | 70.32 | 63.85 | 91.70 | 80.58 |
| SISA | 1 | 78.02 | - | 95.41 | 83.06 | 58.26 | - | 86.90 | 79.42 | 80.14 | - | 84.91 | 75.47 |
| | 2 | 77.72 | 78.17 | 95.08 | 81.92 | 56.47 | 57.97 | 85.02 | 76.62 | 78.85 | 78.64 | 83.53 | 73.58 |
| | 3 | 75.82 | 77.64 | 92.98 | 81.50 | 55.14 | 57.07 | 84.12 | 76.45 | 78.10 | 78.56 | 83.53 | 73.33 |
| | 4 | 75.63 | 76.87 | 92.80 | 80.34 | 54.58 | 56.96 | 83.86 | 76.16 | 77.33 | 77.60 | 82.33 | 72.65 |
| | 5 | 72.91 | 75.86 | 91.58 | 78.82 | 51.90 | 55.67 | 81.76 | 74.30 | 74.83 | 77.09 | 80.05 | 70.92 |
| INF | 1 | 95.15 | - | 97.10 | 78.18 | 66.30 | - | 71.97 | 63.42 | 91.70 | - | 94.46 | 83.20 |
| | 2 | 91.71 | 92.08 | 96.90 | 75.69 | 65.51 | 66.55 | 70.36 | 62.13 | 91.43 | 90.91 | 92.68 | 82.33 |
| | 3 | 91.21 | 91.64 | 96.51 | 75.53 | 65.44 | 66.53 | 70.01 | 61.42 | 90.72 | 90.84 | 92.51 | 81.75 |
| | 4 | 91.05 | 91.60 | 94.73 | 74.52 | 65.40 | 65.55 | 69.59 | 61.32 | 89.48 | 90.03 | 91.97 | 80.96 |
| | 5 | 89.53 | 90.71 | 94.01 | 72.92 | 62.70 | 64.63 | 68.37 | 60.25 | 88.22 | 88.99 | 90.92 | 80.70 |
| IUPC | 1 | 75.25 | - | 88.39 | 78.28 | 56.81 | - | 70.68 | 61.26 | 79.90 | - | 79.62 | 72.35 |
| | 2 | 70.14 | 73.50 | 86.35 | 76.32 | 52.56 | 55.13 | 68.23 | 57.38 | 76.77 | 78.36 | 78.75 | 71.54 |
| | 3 | 69.90 | 73.09 | 85.71 | 75.79 | 51.93 | 55.02 | 66.80 | 57.15 | 75.91 | 77.36 | 77.44 | 69.69 |
| | 4 | 67.65 | 71.84 | 85.42 | 75.51 | 51.54 | 53.91 | 66.69 | 57.08 | 74.78 | 77.16 | 76.81 | 68.39 |
| | 5 | 66.41 | 69.14 | 82.73 | 73.50 | 47.31 | 50.57 | 64.55 | 54.53 | 71.43 | 74.94 | 74.69 | 67.63 |
| *Ours* | 1 | 67.68 | - | 90.05 | 78.60 | 49.81 | - | 71.75 | 63.49 | 76.22 | - | 84.92 | 76.08 |
| | 2 | 66.39 | 65.97 | 89.22 | 77.08 | 48.00 | 44.14 | 71.48 | 61.93 | 75.67 | 74.02 | 83.93 | 74.69 |
| | 3 | 65.90 | 63.52 | 88.78 | 76.60 | 46.50 | 43.75 | 70.58 | 61.73 | 74.84 | 73.04 | 83.61 | 74.46 |
| | 4 | 64.79 | 63.29 | 88.16 | 76.03 | 46.17 | 43.35 | 69.89 | 61.70 | 73.78 | 72.91 | 82.95 | 74.31 |
| | 5 | 63.24 | 63.21 | 87.74 | 74.74 | 43.52 | 41.63 | 69.60 | 60.74 | 70.38 | 71.31 | 82.09 | 73.15 |

Table 9: The relationship between the results on different datasets and the effects of sequential unlearning when the model is VGG.

| Method | Round ($T$) | CIFAR-10 | | | | CIFAR-100 | | | | Fashion-MNIST | | | |
|---|---|---|---|---|---|---|---|---|---|---|---|---|---|
| | | $Acc_F(\downarrow)$ | $Acc_P(\downarrow)$ | $Acc_R(\uparrow)$ | $Acc_T(\uparrow)$ | $Acc_F(\downarrow)$ | $Acc_P(\downarrow)$ | $Acc_R(\uparrow)$ | $Acc_T(\uparrow)$ | $Acc_F(\downarrow)$ | $Acc_P(\downarrow)$ | $Acc_R(\uparrow)$ | $Acc_T(\uparrow)$ |
| Original | - | 95.77 | 94.15 | 95.73 | 85.56 | 96.26 | 95.07 | 94.57 | 87.16 | 97.67 | 98.06 | 97.14 | 90.67 |
| Retrain | 1 | 57.33 | - | 99.07 | 80.99 | 48.50 | - | 81.03 | 69.89 | 82.60 | - | 99.14 | 88.98 |
| | 2 | 52.22 | 51.63 | 96.76 | 78.53 | 46.76 | 41.55 | 79.30 | 68.67 | 77.96 | 77.34 | 97.32 | 86.39 |
| | 3 | 52.20 | 50.69 | 95.97 | 78.45 | 46.62 | 41.36 | 79.08 | 67.80 | 77.45 | 75.71 | 96.97 | 86.38 |
| | 4 | 51.69 | 50.03 | 95.49 | 77.48 | 44.18 | 40.39 | 78.38 | 67.63 | 75.37 | 75.68 | 96.89 | 85.87 |
| | 5 | 48.61 | 46.96 | 93.35 | 75.25 | 40.93 | 38.97 | 76.41 | 65.05 | 73.97 | 72.83 | 94.83 | 83.63 |
| SISA | 1 | 69.75 | - | 89.08 | 67.20 | 64.95 | - | 69.03 | 65.90 | 85.41 | - | 87.96 | 73.08 |
| | 2 | 66.99 | 78.51 | 86.62 | 65.89 | 62.69 | 63.25 | 67.80 | 63.30 | 83.04 | 83.89 | 85.52 | 70.30 |
| | 3 | 66.67 | 77.92 | 86.57 | 65.27 | 61.73 | 62.35 | 67.07 | 63.07 | 81.52 | 83.32 | 84.48 | 69.93 |
| | 4 | 66.35 | 77.90 | 85.82 | 64.30 | 60.32 | 62.34 | 66.77 | 62.69 | 81.45 | 83.01 | 84.48 | 69.67 |
| | 5 | 64.79 | 76.82 | 83.61 | 62.99 | 59.67 | 60.02 | 66.43 | 60.95 | 80.13 | 80.82 | 82.76 | 67.73 |
| INF | 1 | 75.30 | - | 90.48 | 69.99 | 78.25 | - | 80.45 | 67.33 | 82.75 | - | 87.25 | 75.16 |
| | 2 | 75.23 | 75.74 | 89.63 | 69.32 | 75.81 | 76.35 | 79.69 | 65.97 | 80.88 | 78.40 | 86.48 | 73.29 |
| | 3 | 73.05 | 75.73 | 89.45 | 68.98 | 74.99 | 76.33 | 79.62 | 65.63 | 79.66 | 78.24 | 86.30 | 73.16 |
| | 4 | 72.45 | 75.29 | 89.39 | 68.41 | 74.95 | 75.72 | 79.49 | 65.41 | 78.80 | 78.14 | 86.16 | 72.93 |
| | 5 | 71.62 | 74.00 | 87.45 | 67.26 | 72.64 | 74.51 | 77.55 | 64.11 | 78.26 | 76.52 | 84.20 | 71.26 |
| IUPC | 1 | 62.25 | - | 76.37 | 62.52 | 58.78 | - | 63.29 | 59.08 | 88.04 | - | 74.24 | 59.80 |
| | 2 | 60.03 | 62.00 | 74.42 | 61.06 | 56.24 | 59.24 | 61.77 | 57.81 | 83.49 | 86.29 | 71.80 | 57.03 |
| | 3 | 58.65 | 61.96 | 73.37 | 60.42 | 55.82 | 58.88 | 61.55 | 57.56 | 82.75 | 85.37 | 70.89 | 56.95 |
| | 4 | 57.84 | 61.39 | 72.69 | 59.32 | 54.68 | 57.68 | 61.24 | 55.85 | 81.36 | 85.10 | 70.63 | 56.92 |
| | 5 | 54.48 | 58.50 | 69.79 | 57.12 | 50.39 | 55.26 | 57.36 | 54.25 | 79.92 | 82.64 | 67.20 | 54.15 |
| *Ours* | 1 | 57.38 | - | 79.91 | 65.66 | 52.05 | - | 73.28 | 61.50 | 81.41 | - | 89.22 | 73.51 |
| | 2 | 54.29 | 52.34 | 79.19 | 64.81 | 48.58 | 46.63 | 71.55 | 60.16 | 79.50 | 77.20 | 87.90 | 72.38 |
| | 3 | 53.89 | 51.85 | 78.62 | 64.79 | 48.01 | 46.42 | 70.98 | 60.02 | 78.35 | 76.43 | 87.49 | 72.13 |
| | 4 | 52.78 | 51.70 | 78.42 | 64.60 | 47.28 | 45.65 | 70.55 | 58.97 | 78.03 | 76.06 | 87.16 | 71.88 |
| | 5 | 50.87 | 50.00 | 76.91 | 63.14 | 45.94 | 45.29 | 70.33 | 58.70 | 76.10 | 75.59 | 85.68 | 70.35 |

Table 10: The relationship between the results on different datasets and the effects of sequential unlearning when the model is ViT.

| Method | Round (T) | CIFAR-10 | | | | CIFAR-100 | | | | Fashion-MNIST | | | |
|---|---|---|---|---|---|---|---|---|---|---|---|---|---|
| | | $Acc_F(\downarrow)$ | $Acc_P(\downarrow)$ | $Acc_R(\uparrow)$ | $Acc_T(\uparrow)$ | $Acc_F(\downarrow)$ | $Acc_P(\downarrow)$ | $Acc_R(\uparrow)$ | $Acc_T(\uparrow)$ | $Acc_F(\downarrow)$ | $Acc_P(\downarrow)$ | $Acc_R(\uparrow)$ | $Acc_T(\uparrow)$ |
| Original | - | 95.65 | 94.71 | 96.34 | 85.47 | 84.93 | 84.96 | 78.70 | 69.90 | 97.67 | 98.07 | 98.44 | 92.51 |
| Retrain | 1 | 73.55 | - | 96.13 | 80.89 | 33.84 | - | 74.66 | 66.55 | 77.72 | - | 97.46 | 90.04 |
| | 2 | 70.58 | 67.14 | 94.51 | 79.15 | 29.62 | 27.10 | 73.04 | 64.47 | 75.38 | 72.54 | 95.22 | 87.44 |
| | 3 | 65.97 | 66.31 | 93.88 | 78.51 | 28.55 | 25.04 | 72.45 | 64.14 | 73.19 | 71.83 | 94.94 | 86.85 |
| | 4 | 65.61 | 65.50 | 93.80 | 78.48 | 26.51 | 24.68 | 72.15 | 63.67 | 71.53 | 71.43 | 94.83 | 86.24 |
| | 5 | 64.44 | 62.65 | 91.73 | 76.17 | 22.32 | 21.95 | 69.63 | 61.68 | 67.29 | 68.56 | 93.09 | 85.04 |
| SISA | 1 | 79.70 | - | 94.10 | 76.48 | 49.60 | - | 69.94 | 60.72 | 85.41 | - | 96.47 | 84.36 |
| | 2 | 77.90 | 79.99 | 92.10 | 74.57 | 47.20 | 47.95 | 67.63 | 59.66 | 83.00 | 85.42 | 94.16 | 82.47 |
| | 3 | 76.14 | 79.42 | 91.73 | 74.36 | 47.03 | 47.22 | 67.29 | 58.65 | 83.00 | 85.26 | 93.75 | 82.47 |
| | 4 | 75.94 | 79.19 | 91.44 | 73.50 | 47.03 | 46.20 | 66.92 | 58.59 | 82.80 | 82.73 | 93.24 | 82.04 |
| | 5 | 73.17 | 77.64 | 88.97 | 71.84 | 43.10 | 45.94 | 64.48 | 55.79 | 80.59 | 82.25 | 91.03 | 80.17 |
| INF | 1 | 84.39 | - | 93.25 | 77.55 | 57.32 | - | 73.92 | 62.37 | 84.58 | - | 96.30 | 87.41 |
| | 2 | 83.26 | 83.84 | 92.08 | 76.94 | 56.22 | 56.27 | 72.29 | 61.10 | 83.12 | 83.22 | 95.22 | 87.26 |
| | 3 | 81.84 | 83.75 | 91.93 | 76.91 | 55.28 | 56.26 | 72.18 | 60.96 | 80.55 | 82.74 | 94.38 | 86.33 |
| | 4 | 80.15 | 83.66 | 91.57 | 76.68 | 53.68 | 55.73 | 71.40 | 60.65 | 80.29 | 82.44 | 94.19 | 86.06 |
| | 5 | 78.97 | 81.91 | 90.57 | 74.62 | 52.45 | 55.15 | 70.50 | 59.67 | 80.16 | 81.91 | 92.96 | 84.69 |
| IUPC | 1 | 80.00 | - | 92.15 | 78.66 | 54.02 | - | 70.73 | 61.83 | 80.83 | - | 93.59 | 85.09 |
| | 2 | 77.77 | 79.28 | 89.04 | 76.01 | 52.28 | 54.88 | 67.42 | 60.66 | 77.45 | 79.54 | 91.33 | 82.34 |
| | 3 | 77.74 | 78.84 | 87.88 | 74.91 | 51.31 | 53.69 | 67.35 | 59.84 | 76.70 | 79.00 | 91.22 | 81.58 |
| | 4 | 76.81 | 76.39 | 86.99 | 72.82 | 49.80 | 52.19 | 66.26 | 59.76 | 76.56 | 78.51 | 91.16 | 79.55 |
| | 5 | 72.08 | 74.15 | 85.19 | 71.15 | 46.82 | 50.38 | 64.89 | 56.11 | 72.71 | 75.44 | 88.05 | 77.68 |
| *Ours* | 1 | 74.65 | - | 91.68 | 77.03 | 34.53 | - | 70.72 | 62.62 | 77.45 | - | 90.77 | 84.72 |
| | 2 | 72.45 | 70.26 | 90.39 | 75.65 | 33.09 | 29.88 | 69.86 | 61.56 | 75.57 | 72.15 | 89.53 | 84.22 |
| | 3 | 71.56 | 69.95 | 89.91 | 75.58 | 31.73 | 29.87 | 69.83 | 61.48 | 75.15 | 71.58 | 89.41 | 84.02 |
| | 4 | 69.54 | 69.81 | 89.59 | 74.89 | 30.79 | 29.72 | 69.30 | 60.70 | 73.04 | 71.16 | 88.91 | 82.76 |
| | 5 | 67.16 | 68.78 | 88.32 | 74.01 | 27.80 | 28.01 | 67.55 | 59.85 | 72.63 | 69.55 | 87.28 | 81.87 |

Table 11: Detailed experimental performance of different methods under varying numbers of unlearning request rounds, where the recommendation model is ResNet-18 and the dataset is CIFAR-10.

| Rounds (T) | Retrain | | | | SISA | | | | INF | | | | IUPC | | | | *Ours* | | | |
|---|---|---|---|---|---|---|---|---|---|---|---|---|---|---|---|---|---|---|---|---|
| | $Acc_F(\downarrow)$ | $Acc_P(\downarrow)$ | $Acc_R(\uparrow)$ | $Acc_T(\uparrow)$ | $Acc_F(\downarrow)$ | $Acc_P(\downarrow)$ | $Acc_R(\uparrow)$ | $Acc_T(\uparrow)$ | $Acc_F(\downarrow)$ | $Acc_P(\downarrow)$ | $Acc_R(\uparrow)$ | $Acc_T(\uparrow)$ | $Acc_F(\downarrow)$ | $Acc_P(\downarrow)$ | $Acc_R(\uparrow)$ | $Acc_T(\uparrow)$ | $Acc_F(\downarrow)$ | $Acc_P(\downarrow)$ | $Acc_R(\uparrow)$ | $Acc_T(\uparrow)$ |
| 1 | 60.78 | 55.57 | 94.47 | 81.45 | 77.61 | 79.10 | 94.61 | 81.91 | 89.02 | 88.38 | 95.90 | 76.02 | 73.26 | 76.58 | 87.90 | 76.11 | 64.25 | 62.28 | 89.61 | 77.54 |
| 2 | 60.44 | 55.34 | 94.22 | 81.23 | 76.76 | 78.73 | 94.09 | 81.45 | 88.04 | 88.13 | 95.37 | 75.53 | 71.43 | 73.67 | 85.87 | 76.03 | 63.58 | 61.57 | 88.73 | 76.52 |
| 3 | 60.30 | 55.15 | 94.16 | 81.13 | 76.41 | 78.53 | 93.63 | 81.06 | 87.93 | 88.08 | 95.21 | 75.22 | 71.16 | 73.50 | 84.55 | 75.91 | 63.58 | 61.24 | 88.70 | 76.50 |
| 4 | 60.24 | 55.09 | 94.10 | 81.07 | 76.19 | 78.35 | 93.60 | 81.02 | 87.86 | 87.88 | 95.10 | 75.20 | 71.08 | 72.83 | 81.70 | 75.60 | 61.99 | 60.88 | 88.31 | 76.07 |
| 5 | 59.54 | 54.90 | 93.90 | 80.87 | 75.33 | 78.00 | 93.33 | 80.22 | 86.79 | 87.38 | 94.68 | 74.88 | 69.66 | 70.77 | 81.39 | 72.17 | 61.99 | 60.88 | 88.31 | 76.07 |
| 6 | 57.00 | 54.17 | 90.82 | 78.80 | 73.28 | 74.73 | 91.76 | 77.91 | 86.41 | 86.83 | 93.67 | 73.95 | 68.71 | 70.26 | 80.70 | 71.09 | 59.51 | 56.59 | 84.90 | 75.34 |
| 7 | 56.66 | 53.98 | 90.53 | 78.43 | 72.60 | 73.96 | 91.03 | 77.00 | 85.26 | 86.42 | 93.03 | 73.39 | 68.44 | 70.09 | 80.54 | 71.02 | 58.34 | 58.17 | 84.11 | 75.26 |
| 8 | 56.31 | 53.91 | 90.52 | 78.42 | 72.45 | 73.93 | 90.96 | 76.90 | 85.26 | 86.33 | 92.93 | 73.31 | 68.15 | 69.11 | 79.73 | 70.93 | 57.41 | 55.78 | 83.83 | 74.83 |
| 9 | 56.13 | 53.89 | 90.23 | 78.37 | 72.14 | 73.60 | 90.95 | 76.87 | 85.25 | 86.13 | 92.84 | 73.23 | 66.83 | 68.85 | 77.52 | 68.85 | 55.88 | 55.25 | 83.75 | 74.53 |
| 10 | 55.39 | 53.41 | 90.05 | 77.97 | 71.19 | 73.42 | 90.46 | 76.22 | 84.07 | 86.10 | 92.68 | 72.97 | 66.41 | 68.53 | 77.06 | 68.16 | 55.88 | 54.76 | 83.66 | 73.79 |
| 11 | 50.75 | 50.25 | 88.17 | 77.01 | 70.39 | 71.25 | 86.59 | 71.35 | 83.18 | 86.04 | 88.81 | 70.65 | 64.66 | 66.59 | 74.50 | 67.48 | 54.20 | 51.27 | 82.33 | 72.00 |
| 12 | 50.42 | 50.00 | 87.78 | 76.74 | 70.28 | 71.01 | 85.94 | 70.84 | 82.37 | 85.66 | 88.47 | 70.22 | 63.13 | 66.43 | 74.39 | 66.62 | 54.15 | 50.91 | 82.09 | 71.47 |
| 13 | 50.15 | 49.97 | 87.74 | 76.58 | 69.97 | 70.42 | 85.72 | 70.70 | 82.30 | 85.66 | 88.35 | 69.99 | 61.38 | 66.25 | 74.35 | 66.54 | 53.17 | 50.33 | 81.93 | 71.37 |
| 14 | 49.97 | 49.93 | 87.61 | 76.57 | 69.25 | 70.33 | 85.29 | 70.60 | 81.91 | 85.58 | 88.17 | 69.99 | 60.94 | 65.15 | 71.92 | 63.85 | 52.54 | 49.88 | 81.74 | 71.03 |
| 15 | 49.27 | 49.51 | 87.46 | 76.10 | 68.84 | 69.84 | 84.87 | 69.85 | 81.19 | 85.26 | 87.70 | 69.75 | 60.63 | 65.15 | 70.69 | 62.78 | 52.28 | 49.50 | 81.11 | 70.84 |
| 16 | 48.17 | 46.30 | 82.62 | 71.22 | 62.49 | 64.89 | 77.22 | 66.70 | 78.64 | 80.14 | 83.87 | 68.37 | 60.37 | 63.83 | 70.25 | 62.37 | 52.04 | 49.10 | 79.15 | 70.44 |
| 17 | 47.46 | 45.95 | 82.33 | 71.03 | 61.80 | 64.04 | 76.86 | 66.06 | 77.84 | 79.92 | 83.65 | 67.97 | 58.37 | 63.07 | 69.11 | 60.88 | 51.48 | 48.46 | 78.12 | 70.40 |
| 18 | 47.13 | 45.94 | 82.27 | 70.93 | 61.69 | 63.88 | 76.59 | 65.82 | 77.42 | 79.77 | 83.61 | 67.84 | 57.76 | 61.83 | 68.15 | 60.65 | 50.85 | 48.15 | 78.08 | 70.29 |
| 19 | 47.03 | 45.93 | 82.07 | 70.74 | 60.21 | 63.71 | 76.59 | 65.73 | 77.26 | 79.58 | 83.43 | 67.76 | 57.38 | 61.63 | 67.98 | 59.78 | 50.39 | 48.10 | 77.98 | 70.09 |
| 20 | 46.71 | 45.53 | 81.95 | 70.43 | 59.56 | 63.33 | 75.79 | 64.89 | 76.45 | 79.25 | 82.90 | 67.41 | 53.41 | 61.59 | 64.27 | 56.81 | 49.14 | 47.38 | 77.62 | 69.77 |

Table 12: Performance on ImageNet-1K, the best results are highlighted in **bold**.

| Dataset | Method | VGG | | | | Swin Transformer | | | |
|---|---|---|---|---|---|---|---|---|---|
| | | $Acc_F(\downarrow)$ | $Acc_P(\downarrow)$ | $Acc_R(\uparrow)$ | $Acc_T(\uparrow)$ | $Acc_F(\downarrow)$ | $Acc_P(\downarrow)$ | $Acc_R(\uparrow)$ | $Acc_T(\uparrow)$ |
| ImageNet-1K | Original | 98.02 | 91.43 | 97.89 | 96.60 | 98.15 | 98.43 | 97.18 | 95.38 |
| | Retrain | 54.77 | 51.37 | 82.82 | 79.25 | 61.48 | 60.51 | 92.14 | 90.47 |
| | INF | 85.42 | 87.23 | 83.74 | **80.35** | 80.57 | 76.13 | 89.24 | 87.51 |
| | *Ours* | **57.38** | **53.71** | **86.61** | 75.34 | **65.28** | **63.43** | **89.29** | **88.42** |

Table 13: Performance on Swin Transformer, the best results are highlighted in **bold**.

| Model | Method | CIFAR-100 | | | | ImageNet-1K | | | |
|---|---|---|---|---|---|---|---|---|---|
| | | $Acc_F(\downarrow)$ | $Acc_P(\downarrow)$ | $Acc_R(\uparrow)$ | $Acc_T(\uparrow)$ | $Acc_F(\downarrow)$ | $Acc_P(\downarrow)$ | $Acc_R(\uparrow)$ | $Acc_T(\uparrow)$ |
| Swin Transformer | Original | 87.58 | 85.32 | 89.25 | 86.09 | 98.15 | 98.43 | 97.18 | 95.38 |
| | Retrain | 66.83 | 64.73 | 82.66 | 79.99 | 61.48 | 60.51 | 92.14 | 90.47 |
| | INF | 79.52 | 80.77 | **81.32** | **78.42** | 80.57 | 76.13 | 89.24 | 87.51 |
| | *Ours* | **68.35** | **68.93** | 79.36 | 73.72 | **65.28** | **63.43** | **89.29** | **88.42** |

## F.2 Experiments on Larger Datasets and Models

We have conducted additional experiments on the larger ImageNet-1K dataset (Deng et al., 2009) and a transformer-based classification model, Swin Transformer (Liu et al., 2021). The results are presented in Table 12 and Table 13, respectively. These results demonstrate that our method exhibits good applicability across large-scale datasets and diverse model architectures.

## F.3 Different Unlearning Rounds

To comprehensively evaluate the effectiveness of our proposed method, we report the performance of each round in Table 11, with the total unlearning rounds $T = 20$.

## F.4 Hyperparameter

We evaluate the performance of ResNet-18 with various hyperparameter settings on CIFAR-10, CIFAR-100, and Fashion-MNIST to explore the effect of hyperparameter $\alpha$. The results of the hyperparameter experiments on CIFAR-10 are depicted in Figure 4 in the main text. For CIFAR-100 and Fashion-MNIST, the hyperparameter experiments are illustrated in Figures 5.

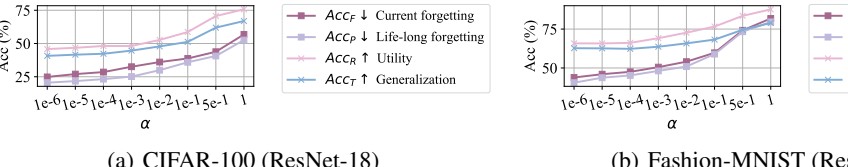

(a) CIFAR-100 (ResNet-18)      (b) Fashion-MNIST (ResNet-18)

Figure 5: Effect of hyperparameter $\alpha$ w.r.t. effectiveness of unlearning on CIFAR-100 and Fashion-MNIST.

In general, we have consistent observations across all three datasets. At $\alpha = 0.5$, the results show that the model achieves an ideal balance between unlearning performance and utility/generalization ability. This suggests that $\alpha = 0.5$ is an appropriate setting that ensures both effective unlearning of the forget set and reliable utility performance on the retain set.

## F.5 Qualitative Analysis

We utilize Centered Kernal Alignment (CKA) (Kornblith et al., 2019) to measure the similarity between the parameters of each layer in two different models. CKA enjoys invariance to invertible linear transformation, orthogonal transformation, and isotropic scaling. This property helps eliminate the randomness and stochastic variations of parameters in deep learning models. Specifically,

CKA takes as input the representation of each layer. Figure 6 displays a heatmap of the CKA correlations between the unlearned model and the retrained model, offering a clear visual comparison. The results indicate that our method is most closely aligned with retraining from scratch at the parameter level. This insight is consistent with the theoretically optimal results described in Section C, where the theoretically optimal model denotes the one obtained through retraining.

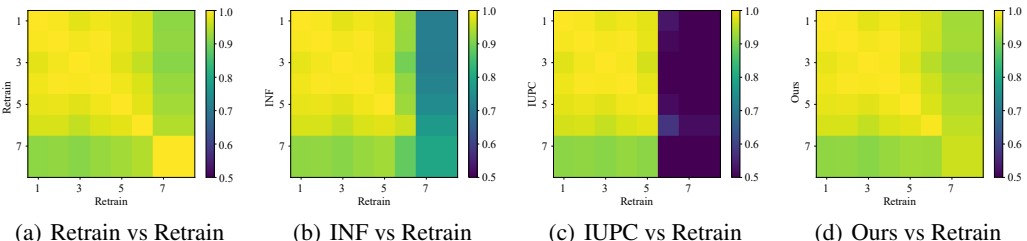

    (a) Retrain vs Retrain     (b) INF vs Retrain     (c) IUPC vs Retrain     (d) Ours vs Retrain

Figure 6: Parameter similarity between unlearned model and Retrain, where the recommendation model is ResNet-18, and the dataset is CIFAR-10. SISA is omitted because it outputs an ensemble system instead of a single model.

### F.6 UNLEARN ALL CLASS SAMPLES IN A SINGLE ROUND

We have supplemented our ResNet-18 experiments with a scenario in which the user selects and unlearns all class samples in a single round. We set the number of unlearning rounds to three, and the unlearning performance at each round is summarized in Table 14. It can be seen that unlearning one class at a time produces a more pronounced and effective unlearning outcome. This is because, in Eq. equation 9, their gradients are highly aligned when the unlearning set comprises samples from only one category. Consequently, the influence-function term accumulates along that category's discriminative direction, rapidly diminishing the model's ability to distinguish that class. Moreover, since the retain set contains no samples of the forgotten class, its gradient contributions toward other categories remain minimal, rendering performance degradation on the remaining classes more controllable. By contrast, a randomly sampled unlearning set typically mixes multiple classes, causing gradient directions to partially cancel out and hindering the complete removal of any single class's discriminative information.

Table 14: Unlearn all class samples in a single round.

| Round | CIFAR-10 | | | | CIFAR-100 | | | | Fashion-MNIST | | | |
|---|---|---|---|---|---|---|---|---|---|---|---|---|
| | $Acc_F(\downarrow)$ | $Acc_P(\downarrow)$ | $Acc_R(\uparrow)$ | $Acc_T(\uparrow)$ | $Acc_F(\downarrow)$ | $Acc_P(\downarrow)$ | $Acc_R(\uparrow)$ | $Acc_T(\uparrow)$ | $Acc_F(\downarrow)$ | $Acc_P(\downarrow)$ | $Acc_R(\uparrow)$ | $Acc_T(\uparrow)$ |
| T=1 | 10.71 | - | 92.71 | 82.61 | 7.14 | - | 93.61 | 85.92 | 14.19 | - | 93.74 | 85.11 |
| T=2 | 8.93 | 10.12 | 90.39 | 80.61 | 7.02 | 6.81 | 91.73 | 83.93 | 12.49 | 13.47 | 93.01 | 83.54 |
| T=3 | 8.87 | 9.37 | 89.12 | 79.61 | 6.79 | 6.35 | 90.26 | 81.92 | 12.31 | 13.08 | 92.16 | 81.32 |

