# OpenReview forum: "Ensuring Life-long Forgetting in Sequential Unlearning via Source-free Optimization"
_ICLR.cc/2026/Conference — ICLR 2026 Conference Withdrawn Submission_

### Official Review · Reviewer_DCSm · 2025-10-30

**Soundness:** 3
**Presentation:** 3
**Contribution:** 3
**Rating:** 6
**Confidence:** 4

**Summary:**

This paper investigates the sequential unlearning setting, where data requiring removal arrives in a stream and unlearning must be performed immediately upon arrival, in contrast to the conventional batch unlearning framework. The paper first applies existing unlearning techniques to this sequential scenario and empirically demonstrates substantial performance degradation. To address this limitation, the paper introduces a new method specifically tailored for sequential unlearning. The proposed approach incorporates three key strategies: (1) employing pseudo labels for the forget set to transform a maximization objective into a minimization one; (2) introducing a source-free optimization term to approximate prior forgetting loss without requiring original forget data; and (3) leveraging influence functions to improve computational efficiency. A series of experiments validates the effectiveness of the proposed approach.

**Strengths:**

1. The paper clearly identifies and motivates the need for new unlearning techniques in the sequential unlearning setting.
2. The proposed method achieves good performance in both unlearning effectiveness and computational efficiency.
3. Several experiments and ablation studies are conducted to substantiate the contributions.
4. The manuscript is well structured, and the presentation is clear and readable.

**Weaknesses:**

1. The experiments lack multiple independent runs, with the mean and standard deviation reported.
2. The optimization procedure appears to store the model parameters and gradients from the previous round. Such two-version updates may unintentionally leak information about the data to be forgotten, as highlighted in prior work [1].
3. For the evaluation metric Membership Inference Attacks (MIAs), more powerful and recent techniques, such as LiRA-based MIAs, could be considered for a more accurate assessment.

[1] Chen, Min, et al. "When machine unlearning jeopardizes privacy." Proceedings of the 2021 ACM SIGSAC conference on computer and communications security. 2021.

**Questions:**

1. How many samples are in the forget set per round, and how sensitive is the unlearning performance to its size?
2. Does the order in which data arrives for unlearning affect the resulting unlearning performance?
3. Have you compared performance with vs. without the influence-function component?

---

### Official Review · Reviewer_jEMj · 2025-10-31

**Soundness:** 3
**Presentation:** 3
**Contribution:** 4
**Rating:** 6
**Confidence:** 4

**Summary:**

This paper proposes a novel approach for sequential machine unlearning, addressing two major challenges of existing batch unlearning methods, first, the lack of life-long forgetting, where previously forgotten data can be unintentionally relearned, and second, inefficiency caused by repeated computation over retain sets or inaccessible past forget sets.

The authors introduce a new approach  that adds a life-long forgetting term to the unlearning objective and converts the loss maximization problem into a stable minimization using pseudo labels. They further propose a source-free optimization strategy that relies only on model parameters and derived loss bounds instead of using the original data. The method is evaluated on CIFAR-10, CIFAR-100, and Fashion-MNIST using models like ResNet-18, VGG, and ViT, and compared with several baselines such as SISA, INF, and IUPC.
Results show that the proposed method achieves better life-long forgetting, competitive accuracy on retain and test sets, and significantly higher efficiency.

**Strengths:**

The paper tackles an important and relatively under-explored problem in the field of machine unlearning. It focuses on the practical challenge of sequential unlearning, where data removal requests appear one after another instead of all at once. By addressing this setting, the work contributes to making unlearning methods more realistic and applicable to real-world scenarios.

One of the key ideas introduced in this work is the concept of life-long forgetting. This ensures that any data unlearned in earlier rounds remains forgotten even as the model continues to process new information. Such a mechanism is essential for maintaining consistent and reliable unlearning performance over time.

The paper also proposes a source-free optimization strategy that removes the dependence on retain sets or previously forgotten data. This design makes the approach far more efficient, as it eliminates the need to repeatedly access large datasets or store sensitive information from previous unlearning rounds.

The experimental setup in this work is well designed and comprehensive.  The study combines theoretical insights with comprehensive evaluations, demonstrating consistent forgetting performance across multiple datasets and architectures. The experimental design is clear, appropriate, and reproducible. The results consistently show improvements in both unlearning effectiveness and computational efficiency compared to existing methods. This demonstrates that the proposed method is both theoretically sound and practically effective.
The inclusion of detailed ablation and hyperparameter studies strengthens the paper further. These analyses provide valuable insights into how different components of the proposed method contribute to its performance and help validate the design choices made by the authors.

The paper is clear and well organized. The paper is technically sound, with well motivated objectives and logically structured derivations. The motivation for sequential unlearning and life-long forgetting is presented logically, and the flow between problem formulation, methodology, and results is coherent. The figures and tables effectively summarize the main findings, although some notations and equations are dense and could benefit from improved readability and clearer legends.

**Weaknesses:**

The paper briefly mentions related works in unlearning but does not sufficiently discuss closely relevant studies such as Descent-to-Delete: Gradient-Based Methods for Machine Unlearning (Neel et al., 2021), Machine Unlearning in Forgettability Sequence (Chen et al., 2024), and System-Aware Unlearning Algorithms: Use Lesser, Forget Faster (Lu et al., ICML 2025). Including these would provide better context and highlight the novelty of the proposed framework.

The paper does not examine whether the performance of the proposed method depends on the order in which unlearning requests are processed. No experiments or analyses are provided to study the sensitivity of forgetting quality or model utility to different unlearning sequences. Evaluating this aspect would offer deeper insight into the method’s robustness in real-world scenarios where deletion requests may arrive in varying orders

The theoretical analysis presented in the paper is solid and well connected to the proposed methodology. However, it could be made stronger by including formal convergence proofs or explicit mathematical bounds that quantify the guarantees of life-long forgetting. This would provide a deeper theoretical foundation and further increase confidence in the method’s reliability.

Some of the notations and equations in the paper are quite dense, which may make them difficult to follow for readers who are not experts in optimization or unlearning theory. The clarity of the presentation could be improved by simplifying certain expressions and adding clearer legends, labels, and explanations to the figures.

The paper also offers limited discussion on how the proposed approach would perform in large-scale or real-time deployment settings. It would be helpful to include more details about practical considerations such as memory usage, model size growth, and the latency involved in handling unlearning requests in real-world systems.

 Additionally, while the paper includes membership inference attack (MIA) evaluation in Table 3, it does not examine jailbreak or extraction attacks, which are also important for assessing whether forgotten information can still be recovered.

**Questions:**

Have you explored adaptive weighting for α instead of using a fixed value (α = 0.5) to dynamically balance forgetting and model utility? Insights into whether a data-dependent or round-specific α improves stability would be valuable.

Could you provide more detail on the pseudo-label generation strategy for the forget set and how it influences the results? The paper mentions experimenting with different selection approaches but does not specify which performed best. Understanding whether using random, “least likely,” or other pseudo-labels changes the forgetting–utility trade-off would strengthen the interpretation of results.
How does the model handle contradictory or overlapping unlearning requests where certain samples share semantic similarity with retained data? Clarifying this would help assess the robustness of forgetting in real-world scenarios.

How sensitive is the method to the pre-computed Hessian matrix — does its approximation affect forgetting quality over multiple unlearning rounds? It would be helpful to understand whether these approximations accumulate error or remain stable as the number of sequential updates increases.

Could the proposed source-free optimization framework be extended to multimodal unlearning tasks, such as text–image pairs, to demonstrate broader applicability beyond image classification?

---

### Official Review · Reviewer_75ah · 2025-11-01

**Soundness:** 1
**Presentation:** 2
**Contribution:** 2
**Rating:** 2
**Confidence:** 4

**Summary:**

The paper proposes a sequential, source-free machine unlearning method that keeps every previously forgotten sample forgotten by adding a lifelong-forgetting term, and updates parameters via a Hessian-based upper bound, so each round only uses the current forget set, gaining big efficiency.

**Strengths:**

1. This paper tackles a realistic setting: unlearning requests arrive over time, not in one batch.
2. This paper identifies the “life-long forgetting” issue, where earlier-forgotten data must stay forgotten.

**Weaknesses:**

1. Many basic definitions and unlearning settings are wrong in this paper:
(a) The drop in accuracy does not mean forgetting. Therefore, the improvement of forgetting data accuracy does not mean the re-learning of forgetting data in Figure 1, and lower forgetting data accuracy does not mean better forgetting in the experiment sections
(b) The maximization of forgetting data loss in Eq. 1 should not be the objective for unlearning.
2. The derivation from Eq. 7 to Eq. 8 is not clear.
3. The datasets in the experiments are too small. At least Tiny-Imagenet should be used. In addition, the scales of the models do not match the scales of the dataset--using the Vit model on the Fashion dataset may lead to heavy overfitting.
4. The lower Acc F and higher Acc R and Acc T do not mean higher performance in Table 1
5. The experiments lack reports for std.
6. Few meaningful and SOTA methods are tried as baselines.

**Questions:**

Please refer to the weaknesses

---

### Official Review · Reviewer_MF39 · 2025-11-01

**Soundness:** 2
**Presentation:** 3
**Contribution:** 2
**Rating:** 4
**Confidence:** 2

**Summary:**

The paper studies sequential machine unlearning and argues that batch unlearning methods are not suitable for real deployments where deletion requests arrive over time. It identifies two main challegnes: 1. lack of life-long forgeting (earlier forgotten data can be rememorized in later rounds) 2. inefficiency (methods repeatedly require access to large retain sets or expensive influence estimates). Thus the paper propose a source-free sequential unlearning framework to achieve sequential unlearning while remains efficient. Experiments on CIFAR-10/100 and Fashion-MNIST with ResNet-18, VGG, and ViT compare against multiple baselines are provided to show the effectiveness of the proposed.

**Strengths:**

1. The life-long forgetting is an important challenge, the motviation of the paper is clear.
2. The writing is clear and easy-to-follow, with the design choices are discussed and justified.
3. Experiments on multiple datasets, comparing with several baselines are provided, with ablation study, to show the effectiveness of the proposed.

**Weaknesses:**

1. Datasets: The datasets used in the experiments are limited to small datasets (cifar10, cifar100 and fashion-minist). Since the ViT model is also included, providing results on larger datasets would enhance the experiment part of the paper.
2. It seems that in this paper, worse performance on the forget data (lower AccF) is explicitly interpreted as better unlearning, which is not necessarily true. Therefore, the discussion of the experimental results should be revised.
3. Additional metrics such as the forgetting–retention gap should be included to better support the claims of the paper.
4. The size of the forget set seems fixed. Providing results on varied size of forget set would enhance the paper.

**Questions:**

1. Have the authors compared with a retrained model to ensure that both behave similarly on the forget data?
2. The experiments are limited to image classification tasks on small benchmarks. Could the proposed method generalize to larger datasets or other domains (e.g., text datasets)? Providing results on larger datasets and insights on how it would handle non-vision data would strengthen the paper.
3. The forget set size per round appears to be fixed throughout the experiments. Have the authors explored varying the size or composition? Such analysis would help understand how the method scales under different unlearning intensities.

---

### Note · Authors · 2025-11-25

I have read and agree with the venue's withdrawal policy on behalf of myself and my co-authors.